

# Effect of nonlinear tide-surge interaction in the Pearl
# River Estuary during Typhoon Nida (2016)
Linxu Huang[1,2], Tianyu Zhang[2,3,4], Shouwen Zhang[2*], Hui Wang[1,5]
1.  Institute of Marine Science and Technology, Shandong University, Qingdao 266237, China
2. Southern Marine Science and Engineering Guangdong Laboratory (Zhuhai), Zhuhai 519082, China
3. Laboratory for Coastal Ocean Variation and Disaster Prediction, College of Ocean and Meteorology, Guangdong
Ocean University, Zhanjiang 524088, China
4. Key Laboratory of Climate, Resources and Environment in Continental Shelf Sea and Deep Sea of Department
of Education of Guangdong Province, Guangdong Ocean University, Zhanjiang 524088, China
5. National Marine Environmental Forecasting Center, Beijing 100086, China
*zhangshouwen@sml-zhuhai.cn
## Abstract
Storm surge is one of the most significant marine dynamic disasters affecting the coastal
areas worldwide. An in-depth study of its mechanisms is crucial for improving forecasting skills
and implementing better prevention measures. In this study, a numerical model based on the
Advanced Circulation Model (ADCIRC) was used to investigate the characteristics of storm
surges and the mechanisms of tide-surge interaction in the Pearl River Estuary (PRE) during
Typhoon Nida (2016). Three different types of model runs were conducted to distinguish water
level variations caused by astronomical tides, storm surges, and tide-surge interactions. The results
indicated that storm surges are primarily modulated by tides through tide-surge interactions. The
nonlinear effect is mainly generated by the nonlinear local acceleration term and convection term
from the tide-surge interactions in the study area. However, they are predominantly governed by
the nonlinear wind stress term and bottom friction term in shallow water regions such as the
northern part of Qi'ao Island and Shenzhen Bay. Additionally, variations in the y component of the
nonlinear momentum terms are more significant than those in the x component. To investigate the
impact of tidal phase on surge response to Typhoon Nida, we altered the landfall timing to
introduce variations in PRE characteristics. The results shows that the contribution ratio of each
nonlinear term changes little, their magnitudes fluctuate depending on the timing of landfall.
**Keywords:** Tide-surge interaction; Storm surge; ADCIRC; Nonlinear effect

## 1. Introduction
Storm surges are abnormal rises in sea level caused by atmospheric pressure and wind stress,
such as tropical cyclones (TCs) and fronts. TCs, also known as typhoons or hurricanes, can induce
storm surges with extreme water levels that result in severe economic losses and human casualties
in coastal areas, especially when they coincide with high astronomical tides (Flather, 1994). The
conventional method for forecasting storm surges during typhoons involves predicting the water
level under a specified wind field and then linearly add that water level to the predicted
astronomical tides (Heaps, 1983). However, studies have demonstrated that the effect of tide-surge
interaction is nonlinear (Johns et al., 1985; Bernier and Thompson, 2007; Quinn et al., 2012).



Compared with observations, the simple linear superposition of astronomical tides and separately
calculated surges can lead to errors of up to 1-2 m (Rego and Li, 2010).
It is widely known that the total water level can be divided into three main components: the
astronomical tide, the pure storm surge caused by atmospheric forcing, and the nonlinear residual
level caused by tide-surge interaction which is a significant source of error in storm surge
prediction (Idier et al., 2012; Xu et al., 2016; Yang et al., 2019). Observations and simulations
have indicated that storm surges are influenced by astronomical tides, and the effect of nonlinear
tide-surge interaction can significantly modulate water levels in shallow regions. There are two
main characteristics of storm surges due to the nonlinear tide-surge interaction. One characteristic
is that the peak storm surge height near high tide is typically lower than that near low tide, which
could increase surge levels during the rising tide and decrease them at high tide (Rossiter, 1961;
Wolf, 1978; Horsburgh and Wilson, 2007; Olbert et al., 2013). Another characteristic is the
variation in storm surge intensity, where the surge is notably stronger during low tide compared to
high tide (Horsburgh and Wilson, 2007; Feng et al., 2016; Song et al., 2020). The tide-surge
interaction comprises three nonlinear physical processes: (a) the nonlinear advective effect from
the advective terms in the momentum equations; (b) the nonlinear bottom friction effect with
quadratic parameterization; (c) the shallow water effect arising from the nonlinear terms related to
the total water depth in both the mass conservation equation and the momentum equations (Zhang
et al., 2010; Song et al., 2020; Zheng et al., 2020). Zhang et al. (2010) found that bottom friction
was the principal contributor to tide-surge interaction in the Taiwan Strait. Rego and Li (2010)
studied the storm surge induced by Hurricane Rita revealed that the advection terms were
dominant over bottom friction with significant spatial-temporal variations in the nonlinear terms.
In strong current regions, the nonlinear advection term may also play a key role in the dynamics of
nonlinear tide-surge interactions (Wolf, 1978; Rego and Li, 2010; Yang et al., 2019; Hu et al.,
2023). Valle‑Levinson et al. (2013) found that Coriolis accelerations and local accelerations due to
alongshore current may significantly influence the tidal modulation of storm surges.
The characteristics of storm surges and nonlinear effects in the Pearl River Estuary (PRE) are
especially complex, as the PRE is one of the most important economic regions of China. The
topography of the PRE consists of deep channels, shallow shoals, and tidal flats, which makes the
PRE extremely vulnerable to storm surges induced by strong TCs (Zheng et al., 2020). Besides, it
is a typical area with irregular semi-diurnal tides, complicating tide-surge interactions. However,
specific investigations concerning the variability of typhoon landfall timing to tide-surge
interactions and their impact on the temporal and spatial distribution of storm surges within the
PRE are still scarce. Due to the strong tidal dynamics and complex topography in the region,
tide-surge interactions along the PRE are significant, and the mechanisms are complex, which
motivates this work. Therefore, efficient and accurate marine forecasting through the accurate
modeling of storm surges induced by typhoons is essential for mitigating typhoon-induced
disasters in coastal regions.
The main objectives of this study are to investigate the nonlinear residual levels associated
with tide-surge interactions caused by Typhoon Nida, which coincided with the astronomical high
tide in the PRE. Additionally, we aim to explore the dynamic mechanisms by establishing
mathematical relationships between these nonlinear levels and the nonlinear dynamic terms. In
this paper, we utilize a recently developed ADCIRC based PRE surge model, which is nested
within the China Sea tide and surge model, to investigate the mechanism of tide-surge interaction.





In order to better characterize these impacts, different contributions to storm surge events can be
calculated separately using nonlinear terms of the two-dimensional theoretical momentum
equations (Yang et al., 2019; Song et al., 2020; Hu et al., 2023).
In this paper, we will outline the characteristics of Typhoon Nida and provide a detailed
description of the coupled tide-surge model, which is introduced and validated in Section 2. This
is followed by an examination of the distribution of storm surge levels and nonlinear levels, along
with a discussion of the results in Section 3. The conclusions derived from this study are detailed
in Section 4.

## 94 2. Materials and Method

In this study, a coupled tide-surge model was built for Typhoon Nida. The typhoon and
associated numerical model are introduced, and the model setup and validations are also
described.
2.1 Typhoon NIDA
Typhoon Nida generated in the western North Pacific Ocean on 29 July 2016 and began to
move westward rapidly. As shown in Fig 1a, Typhoon Nida passed across the Philippines and
entered the South China Sea (SCS) on July 31, 2016. It then proceeded westward and made
landfall at 19:30 on August 1 in Shenzhen, Guangdong Province, China. The typhoon had a
central pressure of 970 hPa and maximum wind speed exceeding 42 m/s. It is significant that the
peak surge-induced water increase caused by Typhoon Nida coincided with the higher high water
(HHW) tidal phase. Typhoon Nida caused severe economic loss, with an estimated impact of 19
million dollars.

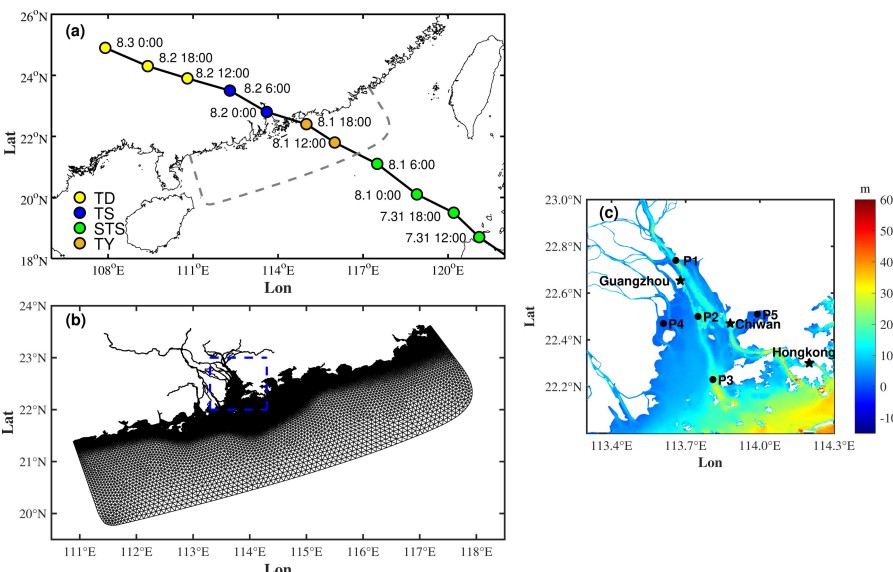


**Figure 1.** (a) The track and intensity of Typhoon Nida; (b) Model domain and grids of the study area;
(c) Location and bathymetry of PRE. Stars represent the tidal gauges, and dots denote the calculation
points of surge levels





2.2 The Numerical Model

Advanced Circulation Model (ADCIRC) was used to simulate the tide and storm surge in

PRE (Luettich et al., 1992). ADCIRC solves the primitive equations with the finite element
method in space and with finite difference method in time (Westerink et al., 1992). Which is
unstructured triangular grids in the horizontal plane to resolve dynamics in complex shorelines .

The basic vertically averaged governing equations, derived from momentum and continuity

are as follows:

$$\frac{\partial H}{\partial t} + \frac{\partial}{\partial x}(UH) + \frac{\partial}{\partial y}(VH) = 0$$

$$\frac{\partial U}{\partial t} + U\frac{\partial}{\partial x}U + V\frac{\partial}{\partial y}U - fV = -g\frac{\partial(\zeta + P_s/g\rho_0)}{\partial x} + \frac{\tau_{sx} - \tau_{bx}}{H\rho_0} - \frac{D_x}{H}$$      (1)

$$\frac{\partial V}{\partial t} + U\frac{\partial}{\partial x}V + V\frac{\partial}{\partial y}V + fU = -g\frac{\partial(\zeta + P_s/g\rho_0)}{\partial y} + \frac{\tau_{sy} - \tau_{by}}{H\rho_0} - \frac{D_y}{H}$$

Where $(U,V)$ are the $x$ and $y$ depth-averaged velocity components; $H=h+\zeta$ is the total water level;
$\zeta$ is free surface elevation; h is the water depth; f is the Coriolis force parameter; g is gravitational
acceleration; $P_s$ is sea surface atmospheric pressure; $\rho_0$ is sea water density; $(\tau_{sx}, \tau_{sy})$ are the $x$ and $y$
components of surface wind stress; $(\tau_{bx}, \tau_{by})$ are the $x$ and $y$ components of bottom friction; $(D_x, D_y)$
are the horizontal momentum diffusion terms.
The surface wind stress parameters $(\tau_{sx}, \tau_{sy})$ are computed as follows:
$$\tau_{sx} = \rho_a C_d W_x \sqrt{W_x^2 + W_y^2}, \tau_{sy} = \rho_a C_d W_y \sqrt{W_x^2 + W_y^2}$$      (2)
Where $\rho_a$ is air density; $(W_x, W_y)$ are the x and y components of wind speed. $C_d$ is the wind drag
coefficient, from Garratt (1977), it is calculated as follows:
$$C_d = 0.001 \times \left(0.75 + 0.067\sqrt{W_x^2 + W_y^2}\right)$$      (3)
The bottom friction $(\tau_{bx}, \tau_{by})$ is computed using the quadratic equation below:
$$\tau_{bx} = \rho_0 C_f U\sqrt{U^2 + V^2}, \tau_{by} = \rho_0 C_f V\sqrt{U^2 + V^2}$$      (4)
The bottom friction drag coefficient $C_f$ is determined by model calibration.

2.3 wind field of typhoon

The wind field model is crucial for accurate storm surge calculations. We employed the

analytical wind model from Holland (1980), which has applied in reconstructing the wind field
during Typhoon Nida. The radial distribution of wind and pressure are determined as follows:
$$P_s(r) = P_c + (P_n - P_c) \cdot \left(-\frac{R_{max}}{r}\right)^B$$      (5)
$$W_g(r) = \sqrt{(P_n - P_c)\frac{B}{\rho_a}\left(\frac{R_{max}}{r}\right)^B \exp\left(-\frac{R_{max}}{r}\right)^B + \left(\frac{rf}{2}\right)^2} - \frac{rf}{2}$$      (6)

Where $r$ is the distance from the typhoon center; $P_n$ is the ambient pressure (1010 hPa) ; $P_c$ is

the central pressure; $R_{max}$ is the maximum wind radius and $W_g$ is wind speed. The $B$ parameter
determines the peak and intensity of the typhoon wind field, and is calculated as follows:



$B = 1.5 + (980 - P_c)/120$                                                 (7)

As $B$ increases, the strong wind becomes increasingly localized near the radius of maximum
winds. For larger $B$, the wind drops off more abruptly both inside and outside the radius of
maximum wind. The $R_{max}$ is calculated as follows:
$R_{max} = 51.6\exp(-0.0223V_{max} + 0.0281\varphi)$                                   (8)

$V_{max}$ is the maximum wind and $\varphi$ is latitude. The inflow angle caused by friction contributes
to wind field asymmetry, and a constant angle of 25 is used in this paper. The central pressure and
position data were retrieved from the China Meteorological Administration (CMA) tropical
cyclone database (Lu et al., 2021).

2.4 Model setting
The grid resolutions were set at a maximum of 600 m at the open boundaries to 100 m within
the PRE region. The domain space was discretized into 325582 triangular cells with 182048 nodes
(Fig 1b). The model utilized mean sea level as its reference datum, and was forced at the open
boundaries by 8 tidal constituents (including $M_2$, $S_2$, $N_2$, $K_2$, $K_1$, $O_1$, $P_1$ and $Q_1$) derived from the
global tidal model TPXO 9 (Egbert and Erofeeva, 2002). The typhoon wind field was generated
using the wind model of Holland (1980). The model was run with a cold start, setting both the
current and water levels to zero at the initial time. The effect of river flow and wind-generated
waves were not considered in our model simulation, as the research is primarily focused on the
tide-surge interactions.
As a semi-enclosed bay, Lingdingyang Bay is regularly affected by both storm surges and
irregular semi-diurnal tides. The trumpet-shaped bay naturally funnels tidal energy, leading to an
amplification of tidal amplitude of at the top of the bay.
First, the definitions of various water levels are explained. The total water elevation ($\zeta_{T+S}$)
can be written as the sum of the tide elevation ($\zeta_T$), pure storm surge elevation ($\zeta_S$) produced by
atmospheric forcing, practical storm surge elevation $\zeta_{PS} = \zeta_{T+S} - \zeta_T$, and the nonlinear residual level
($\zeta_I$) due to tide-surge interaction, such that $\zeta_{T+S} = \zeta_T + \zeta_S + \zeta_I$.
To verify the influence of the tide on storm surges, three simulations were carried out. One
was used to obtain the pure storm surge elevation ($\zeta_S$) by only adding atmospheric forcing, another
was used to obtain the tide elevation ($\zeta_T$) by only adding astronomical tidal forcing, and the other
was used to calculate the total water elevation ($\zeta_{T+S}$) by both atmospheric and tidal forcing.

2.5 Tide and storm surge validation
The correlation coefficient ($R$), root mean square error ($RMSE$), model skill ($Skill$) were used
to validate the computed water level. The definition of the three indicators are determined as
follows:
$R = \dfrac{\sum\limits_{n=1}^{N}\left(M_n - \overline{M_n}\right)\left(C_n - \overline{C_n}\right)}{(N-1)\sigma M \sigma C}$                                      (9)
$RMSE = \sqrt{\dfrac{1}{N}\sum\limits_{n=1}^{N}\left(M_n - C_n\right)^2}$                                         (10)



$$Skill = 1 - \frac{\sum_{n=1}^{N}\left|M_n - C_n\right|^2}{\sum_{n=1}^{N}\left(\left|M_n - \overline{M_n}\right|^2 + \left|C_n - \overline{M_n}\right|^2\right)}$$ (11)
Where $M_n$ and $C_n$ are the measurements and model computed results, respectively, at $N$
discrete point.

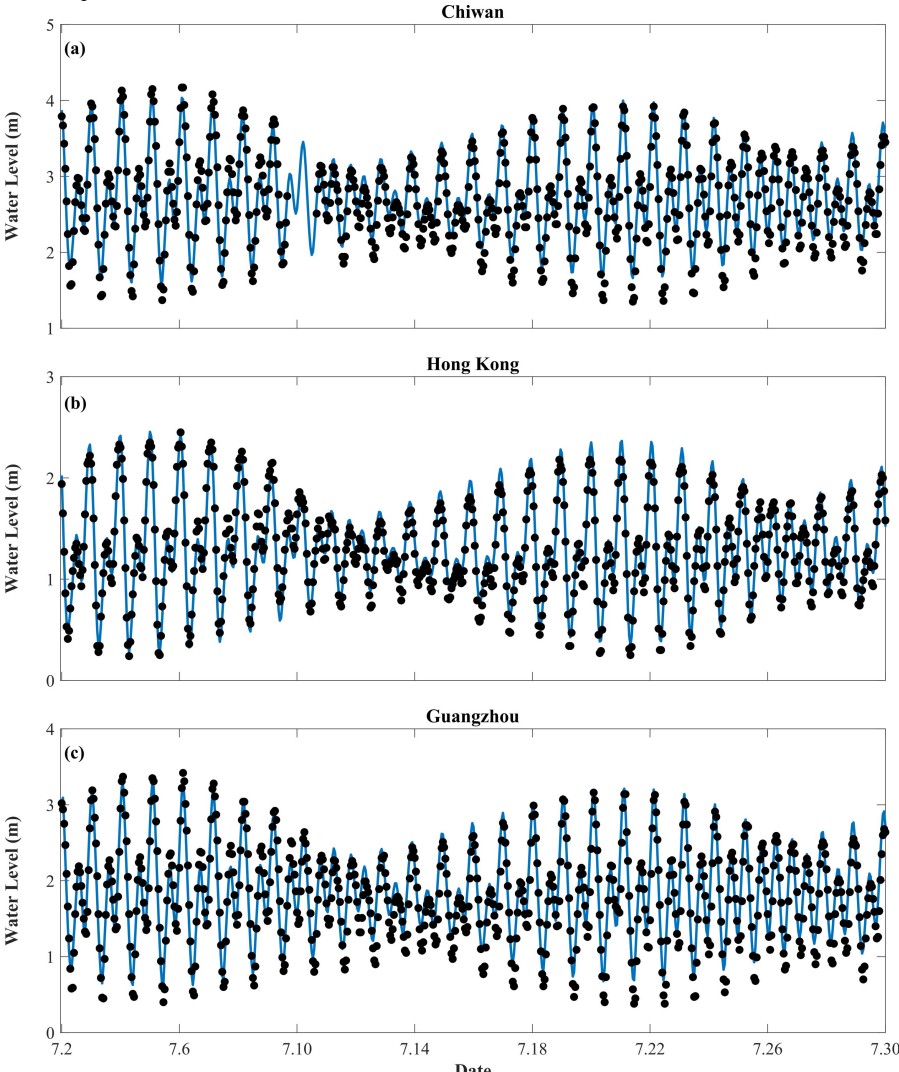

**Figure 2.** Time series comparisons of measured and modeled astronomical tide levels at (a) Chiwan
gauge (b) Hongkong gauge (c) Guangzhou gauge

The computed astronomical tides were initially assessed at three hydrological stations,
Chiwan, Hong Kong, and Guangzhou, during the period from July 2 to July 30, 2016 (Fig 2). The





simulation results demonstrated a close match with the measurements obtained at these three
hydrological stations, as detailed in Table 1. The model predictions exhibited excellent agreement
with the reconstructed astronomical tide, with RMSE values at all three stations are < 0.27 m, both
the R values and Skill values are generally above 0.91.

The model-predicted storm surge levels were further compared with the observed total water
levels at the aforementioned three stations, as depicted in Fig 3. At all three stations mentioned
above, the measured water level data reached its peak (exceeding 2 m) on the evening of August 1,
shortly after Typhoon Nida made landfall. At Chiwan station, the water level exceeded 4 m, with a
positive extreme value error of 0.21 m between the simulated and measured data. At Hong Kong
station, there was 0.13 m error of the positive extreme value between the simulated data and the
measured (Table 1). And at Guangzhou station, there was a positive extreme value error of 0.33 m
between model-predicted storm tide and the measured data.

The numerical results shows that the negative surge levels are overestimated, resulting in
significant errors in storm surge prediction. However, the simulated results for positive surge
levels closely match the observed values, demonstrating that the model used in this study
effectively represents the tidal-surge interactions within the study area.

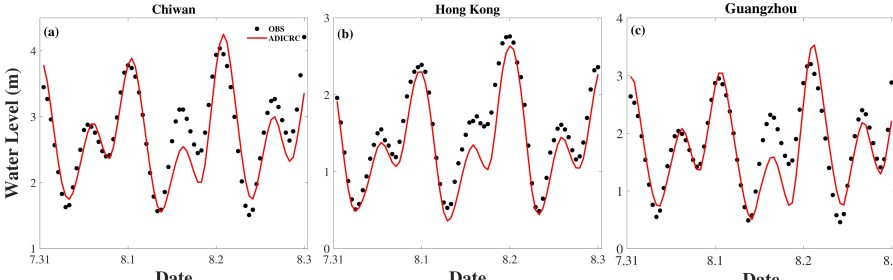

**Figure 3.** Time series comparisons of measured and modeled storm surge levels at (a) Chiwan gauge (b)
Hong Kong gauge (c) Guangzhou gauge

**Table 1.** Comparisons of tide and storm surge between the model simulations and observations

|  | Station | Chiwan | Hong Kong | Guangzhou |
|---|---|---|---|---|
| | R | 0.9470 | 0.9749 | 0.9206 |
| Tide | RMSE (m) | 0.2046 | 0.1047 | 0.2629 |
| | Skill | 0.9368 | 0.9745 | 0.9112 |
| | R | 0.9023 | 0.9728 | 0.8959 |
| Storm Surge | RMSE (m) | 0.3144 | 0.2503 | 0.3467 |
| | Skill | 0.8912 | 0.9169 | 0.8874 |


## 3. Result

### 3.1 Storm surge characteristics of Typhoon Nida

Typhoon Nida passed through the PRE during a spring tide, coinciding with the maximum
positive surge at the HHW tidal phase. Based on this phenomenon, we selected five tidal phases to
investigate the evolution of storm surges before or after Nida made landfall. The spatial and



temporal distribution characteristics of storm surges at different tidal phases in the PRE are
illustrated in Fig 4. Five points are chosen within the internal (P1), middle (P2), external (P3),
northern part of Qi'ao Island (P4), and Shenzhen Bay (P5) regions of PRE to examine this
interaction, as shown in Fig 1c. Notably, the water depth at points P1, P2, and P3 exceeds 10 m,
while the water depth at points P4 and P5 is less than 10 m. At 9:00 on August 1st, during the
lower low water (LLW) tidal phase, the PRE area showed a decreased water level (Fig 4a), while
the nonlinear residual levels were positive in Shenzhen Bay and northern part of Qi'ao Island (Fig
4b). At 15:00 on August 1st, coinciding with the lower high water (LHW) tidal phase, the total
water level in PRE shows a negative to positive trend from northeast to southwest. The most
significant decrease is observed in Shenzhen Bay (Fig 4d). At the same time, the nonlinear
residual levels shows that it is negative in Lingdingyang Bay, except for its top region (Fig 4e). At
19:00 on August 1st, during the higher low water (HLW) tidal phase, the total water level in PRE
area shows a negative trend from northeast to southwest. Notably, Shenzhen Bay experiences the
most significant decrease (Fig 4g). While the nonlinear residual levels are positive, they are
particularly significant in Shenzhen Bay and the northern part of Qi'ao Island (Fig 4h). During the
HHW tidal phase, the total water levels in PRE area show a most substantial increase (Fig 4j).
Conversely, during the same phase, the nonlinear residual levels show a most significant decrease
(Fig 4k). At 10:00 on August 2nd, during the LLW tidal phase, the total water levels in PRE area
are negative, while the nonlinear residual levels are positive.

A comprehensive time series of storm surge elevation ($\zeta_S$), tide elevation ($\zeta_T$), total elevation
($\zeta_{T+S}$), practical storm surge elevation ($\zeta_{PS}$) and nonlinear residuals ($\zeta_{Non}$) for each of the five
points in the study area are also shown in Fig 4. The positive storm surge levels at three points (P1,
P2, and P3) significantly increased from the outer to the inner parts of Lingdingyang Bay. When
the typhoon landfall, the nonlinear residual levels peaked at their maximum positive value and
subsequently reached their maximum negative value before the water level experienced its most
substantial increase. The nonlinear residual levels of P1, P2 and P3 reached their positive maxima
of 0.19 m, 0.14 m and 0.04 m, respectively, which in turn induced effects associated with the
falling tide. The nonlinear residual levels decreased after Typhoon Nida landfall and reached their
negative maxima near the time of the peak positive surge level. Specifically, the negative extreme
value of the nonlinear residual levels of P1, P2 and P3 were -0.43 m, -0.29 m and -0.11 m. The
nonlinear effect within the PRE exhibited a significant increase from its exterior towards the
interior. Additionally, the positive maximum nonlinear residual levels for P4 and P5 were 0.16 m
and 0.14 m respectively, while the negative maximum non-linear residual levels for P4 and P5
were both -0.29 m each. These findings indicate that the overall impact of Typhoon Nida was
characterized by a greater decrease in nonlinear levels than any increase.

We are primarily concerned with the increase in water level that contributes the most to the
total elevation. We calculate the contributions of the pure storm surge elevation, the tide elevation,
the practical surge elevation, and the nonlinear residual to the total elevation from five points in
the PRE region, as shown in Table 2. It is evident from our findings that both pure storm surge and
the tide make positive contributions to the total elevation. In contrast, the nonlinear residuals make
a negative contribution to the total elevation. The practical surge contribution at P1 is found to be
greatest among these five points. Especially, the nonlinear effect at P2 is the most significant when
compared to other points.



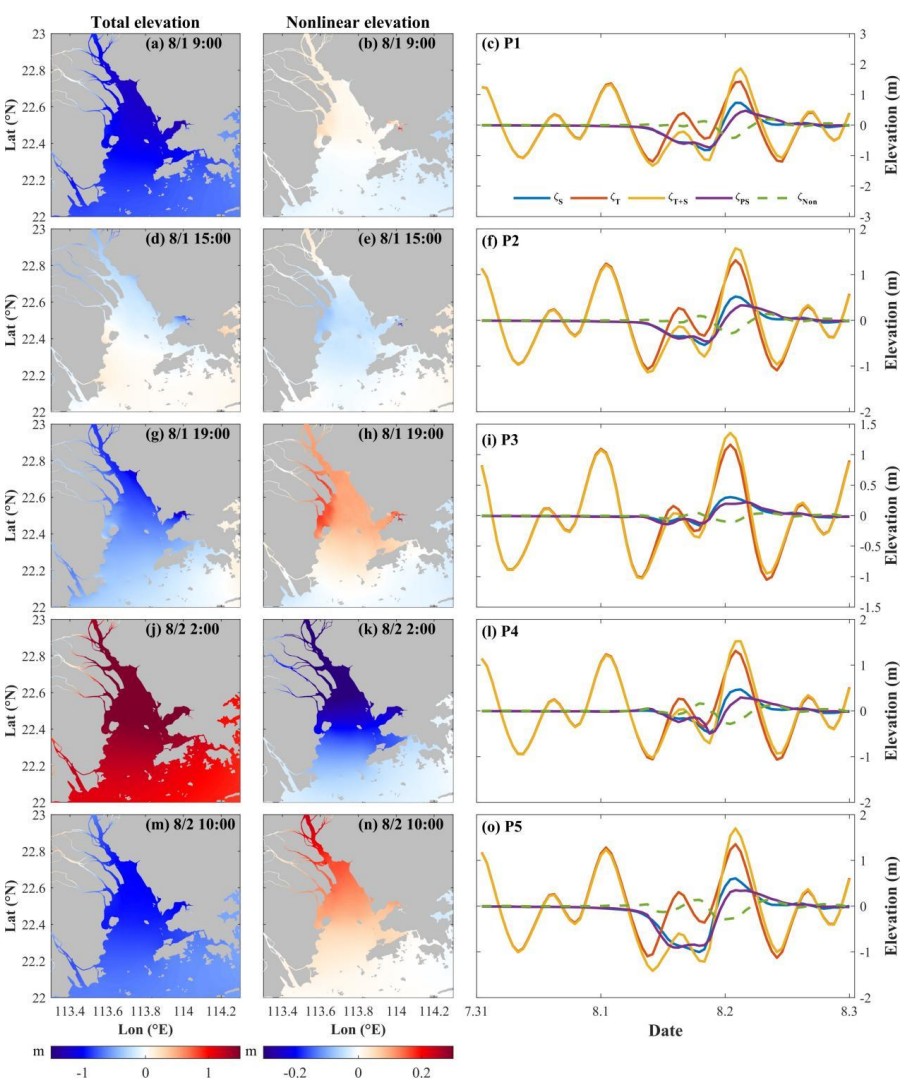

**Figure 4.** Total level (left) and non-linear level (middle) at different tidal phase; Time series of water levels for P1, P2, P3, P4 and P5 locations (right)

To emphasize the importance of the nonlinear effect, we have calculated the ratio of the nonlinear residuals to the pure storm surge and compared it with the ratio of the nonlinear residuals to the practical storm surge, as shown in Table 3. The ratio of the nonlinear residuals to the pure storm surge represents the extent to which the nonlinear effect amplifies or diminishes the direct impact of the storm surge. Meanwhile, the ratio of the nonlinear residuals to the practical storm surge represents how much contribution is made by the nonlinear effect to the practical storm surge. Among all five points, it is observed that at P3, there is a relatively minor contribution in the influence of pure storm surges on water elevation due to nonlinear effects. However, there is an almost 57% negative impact on practical storm surges attributed to these



effects. Notably, at P2, there is a significantly negative contribution from non-linear effects
amounting to nearly 98%. This suggests that compared with other points, P2 experiences a greater
contribution from pure storm surges towards practical storm surges.
**Table 2.** The contribution to the total elevation at its peak moment

| Stations | Surge (%) | Tide (%) | Practical surge (%) | Nonlinear effect (%) |
|----------|-----------|----------|---------------------|----------------------|
| P1 | 39.17 | 76.87 | 23.13 | -16.05 |
| P2 | 33.05 | 83.35 | 16.65 | -16.40 |
| P3 | 22.36 | 85.80 | 14.20 | -8.16 |
| P4 | 30.79 | 80.82 | 19.18 | -11.61 |
| P5 | 35.72 | 79.61 | 20.39 | -15.33 |

**Table 3.** The contribution of the nonlinear effect at the total elevation's peak moment

| Stations | $\zeta_{Non}$ / $\zeta_S$ (%) | $\zeta_{Non}$ / $\zeta_{PS}$ (%) |
|----------|-------------------------------|----------------------------------|
| P1 | –40.97 | –69.38 |
| P2 | –49.61 | –98.47 |
| P3 | –36.49 | –57.46 |
| P4 | –37.71 | –60.54 |
| P5 | –42.91 | –75.17 |


### 3.2 The characteristics of storm surge by different typhoon landfall time

The main causes of tide-surge interactions are the tidal phase alteration caused by surge and
the modulation of surge due to tides (Feng et al., 2019; Zheng et al., 2020). Therefore, we altered
the typhoon landfall times to investigate the characteristics of storm surges during different tidal
phases. The nonlinear effects vary when typhoons make landfall at different tidal phases (Pandey
and Rao, 2019). As shown in Fig 5, the practical storm surge at P1 shows minimal changes with
different typhoon landfall times. When the positive increase in water level caused by pure storm
surge coincides with the HHW tidal phase, the positive extreme values are smallest at each of the
five points. The extreme value of nonlinear level decreases from the inner to the outer regions of
Lingdingyang Bay. However, when the positive increase in water level due to pure storm surge
coincides with HLW tidal phase, which is equivalent to an advance in landfall time by 6 h, it
results in a significantly greater extreme value of the practical surge level compared to others, as
shown in Table 2. Tides make a more significant positive contribution to the total elevation than
pure storm surges, while the nonlinear effect shows a negative contribution (Table 4). When the
positive increase in water level from a pure storm surge coincides with the LHW tidal phase,
corresponding to an 11 h advance in landfall time, the positive extreme value of the practical
storm surge shows little change compared to when it coincides with the HHW tidal phase. Both
surges and tides contribute positively to the total water level. However, the increase in water level
due to the surge is greater than that due to the tide. Consequently, the nonlinear effect shows a
negative contribution leading to a decrease in water level (see Table 4). When the positive increase
in water level from a pure storm surge coincides with LLW tidal phase, occurring at the landfall
time of Typhoon Nida advanced by 16 h, the contribution of surge to the total water level is
negative. Meanwhile, the contributions from the tide and the nonlinear effect are positive (see
Table 4).





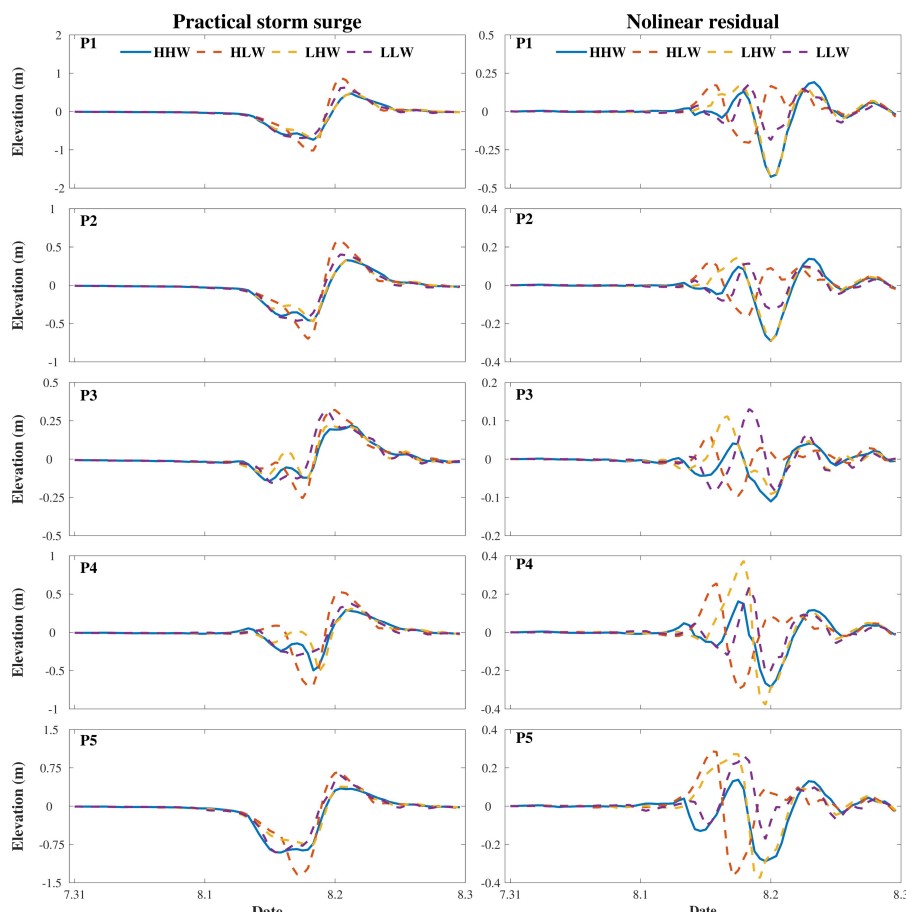

**Figure 5.** Time series of practical storm surge elevation (left) and nonlinear elevation (right) induced by different landfall time forcing at P1, P2, P3, P4 and P5

**Table 4.** Extreme value of practical storm surge elevation caused by different landfall time forcing at P1, P2, P3, P4 and P5

| Stations | Negative extreme value of practical storm surge elevation(m) | | | | Positive extreme value of practical storm surge elevation(m) | | | |
|---|---|---|---|---|---|---|---|---|
| | HHW | HLW | LHW | LLW | HHW | HLW | LHW | LLW |
| P1 | -0.74 | **-1.02** | -0.72 | -0.63 | 0.47 | **0.88** | 0.49 | 0.63 |
| P2 | -0.46 | **-0.70** | -0.47 | -0.46 | 0.33 | **0.59** | 0.35 | 0.40 |
| P3 | -0.14 | **-0.26** | -0.13 | -0.16 | 0.22 | **0.32** | 0.23 | 0.31 |
| P4 | -0.50 | **-0.70** | -0.52 | -0.30 | 0.29 | **0.53** | 0.31 | 0.37 |
| P5 | -0.91 | **-1.34** | -0.79 | -0.90 | 0.35 | **0.67** | 0.38 | 0.60 |

**Table 5.** Extreme value of nonlinear elevation caused by different landfall time forcing at P1, P2, P3, P4 and P5





| Stations | Negative extreme value of nonlinear elevation (m) | | | | Positive extreme value of nonlinear elevation (m) | | | |
|---|---|---|---|---|---|---|---|---|
| | HHW | HLW | LHW | LLW | HHW | HLW | LHW | LLW |
| P1 | -0.43 | -0.20 | -0.43 | -0.19 | 0.19 | 0.17 | 0.17 | 0.18 |
| P2 | -0.29 | -0.16 | -0.29 | -0.13 | 0.14 | 0.13 | 0.15 | 0.11 |
| P3 | -0.11 | -0.10 | -0.09 | -0.09 | 0.04 | 0.06 | 0.11 | 0.13 |
| P4 | -0.29 | -0.3 | -0.38 | -0.21 | 0.16 | 0.26 | 0.37 | 0.23 |
| P5 | -0.29 | -0.36 | -0.37 | -0.17 | 0.14 | 0.29 | 0.27 | 0.26 |


**Table 6.** Contribution of the total elevation when the maximum pure storm surge caused by different
landfall time forcing at P1, P2, P3, P4 and P5

| Stations | Practical surge(%) | | | | Nonlinear effect(%) | | | |
|---|---|---|---|---|---|---|---|---|
| | HHW | HLW | LHW | LLW | HHW | HLW | LHW | LLW |
| P1 | 23.13 | **193.17** | 51.47 | -174.92 | -16.05 | 22.20 | **-36.87** | 27.84 |
| P2 | 16.65 | **224.68** | 49.68 | -68.08 | -16.40 | 25.94 | **-50.12** | 20.09 |
| P3 | 14.20 | **354.88** | 57.33 | -36.91 | -8.16 | 19.24 | **-24.73** | 10.67 |
| P4 | 19.18 | **172.11** | 51.23 | -67.90 | -11.61 | 14.37 | **-35.65** | 25.29 |
| P5 | 20.39 | **207.21** | 55.94 | -133.17 | -15.33 | 18.79 | **-32.97** | 1.06 |


**Table 7.** The importance of the nonlinear effect at P1, P2, P3, P4 and P5 by different landfall time
forcing

| Stations | $\zeta_{Non} / \zeta_S$ (%) | | | | $\zeta_{Non} / \zeta_{PS}$ (%) | | | |
|---|---|---|---|---|---|---|---|---|
| | HHW | HLW | LHW | LLW | HHW | HLW | LHW | LLW |
| P1 | -40.96 | 12.98 | -41.74 | -13.73 | -69.38 | 11.49 | -71.63 | -15.92 |
| P2 | -49.61 | 13.05 | -50.22 | -22.78 | -98.47 | 11.55 | -100.89 | -29.51 |
| P3 | -36.49 | 5.73 | -30.14 | -22.43 | -57.46 | 5.42 | -43.14 | -28.91 |
| P4 | -37.71 | 9.11 | -41.03 | -27.14 | -60.54 | 8.35 | -69.59 | -37.25 |
| P5 | -42.91 | 9.97 | -37.08 | -0.79 | -75.17 | 9.07 | -58.94 | -0.80 |


Above all, the practical storm surge level and the nonlinear level are significantly modulated
by tidal forces, particularly in shallow water areas (Zhang et al., 2017; Zhang et al., 2019; Zhang
et al, 2021). The practical storm surge make the most contribution to the total elevation when the
maximum of the pure storm surge coincides with the HLW tidal phase than other phases. The
nonlinear effect is negative during high tide (HHW and LHW) and positive during low tides
(HLW and LLW) (Horsburgh and Wilson, 2007). The tidal contribution exceeds the surge
contribution when the maximum of pure storm surges coincides with high tides. Conversely, the
contribution of tides is smaller than the surge contribution when the maximum of pure storm
surges coincides with low tides. Notably, when the maximum of pure surges coincides with the
LHW tidal phase, the contribution of the nonlinear effect is the greatest compared to other tidal
phases (see Table 4). When the landfall time coincides with different tidal phases, the extreme
value of positive nonlinear residuals changes little, but the extreme value of negative nonlinear
residuals changes significantly as shown in Table 5. As shown in Table 6, the practical storm surge
makes the greatest contribution when the maximum of the pure storm surge coincides with the



LHW tidal phase, compared to other tidal phases. The nonlinear effect has the greatest impact
when the maximum of the pure storm surge coincides with the HLW tidal phase. When comparing
ratios, it is observed that both the ratio of nonlinear residuals to the pure storm surge and the ratio
of nonlinear residuals to the practical storm surge are positive when their maximum coincides with
HLW tidal phase, while they are negative during other tidal phases (see Table 7). This indicates
that an increase in water elevation due to non-linear effects becomes significant only when their
maximum coincides with HLW tidal phase.

**3.3 Dynamic mechanism of nonlinear residual levels caused by Typhoon Nida**
To further analyze the source of the tide-surge interaction and its nonlinear effects, we
utilized the formula proposed by (Yang et al., 2019) for calculating these nonlinear terms. As
shown in Fig 2, five representative points were selected to illustrate the nonlinear effects of
tide-surge interactions in the PRE.
$$\frac{\partial U_{NS}}{\partial t} + \psi_x(U_{NS}, V_{NS}) - fV_{NS} - \tau_x^S - \tau_x^B = -g\frac{\partial \zeta_I}{\partial x}$$
$$\frac{\partial V_{NS}}{\partial t} + \psi_y(U_{NS}, V_{NS}) + fU_{NS} - \tau_y^S - \tau_y^B = -g\frac{\partial \zeta_I}{\partial y}$$
(12)

The calculated results of nonlinear dynamic terms, including the nonlinear local acceleration
term $\frac{\partial U_{NS}}{\partial t}$ and $\frac{\partial V_{NS}}{\partial t}$, the nonlinear convection term $\psi_x$, $\psi_y$, the nonlinear Coriolis force term
$fU_{NS}$, $fV_{NS}$, the nonlinear wind stress term $\tau_x^S$, $\tau_y^S$, and the nonlinear bottom friction term $\tau_x^B$, $\tau_y^B$ at
each points in $x$ and $y$ direction are shown in Fig 6:
In the eastward direction (x component) at P1, the nonlinear local acceleration term plays a
dominant role, with some contribution from the nonlinear bottom friction term. It is important to
note that this occurs within Lingdingyang Bay. The effects of nonlinear advection and Coriolis
force are deemed negligible. As Typhoon Nida approached Lingdingyang Bay after 18:00 on 1
August 2016, the local acceleration term began to increase and reached a positive maximum at
00:00 on 2 August 2016, similar to the Coriolis term. The advection term reached its positive
maximum one hour later, simultaneously with the negative maximum of the nonlinear residual
water level. Although the wind stress term is minimal, the bottom friction term reached its
negative maximum at 2:00 on 2 August 2016, coinciding with the surge reaching its positive
maximum. In the northward direction (y component), the amplitude of nonlinear local acceleration
remains large; however, it is surpassed by the leading role played by the nonlinear advection term
which reached its positive maximum at 2:00 on 2 August 2016.
In the eastward direction at P2, the values of various nonlinear terms were relatively small,
contributing little to the overall nonlinear effect, with the wind stress term plays a minor role of all
nonlinear terms. However, in the northward direction, the local acceleration term plays a leading
role, the value reached its negative maximum at 22:00 on 1 August 2016, and reached its positive
maximum at 6:00 on 2 August 2016.
In the eastward direction at P3, the nonlinear Coriolis dominated, the values reached its
positive maximum at 23:00 on 1 August 2016. Concurrently, both the nonlinear advection term
and the local acceleration term reached their positive maximums, each making a significantly
contribution. Among all the nonlinear terms, the wind stress plays a minor role.



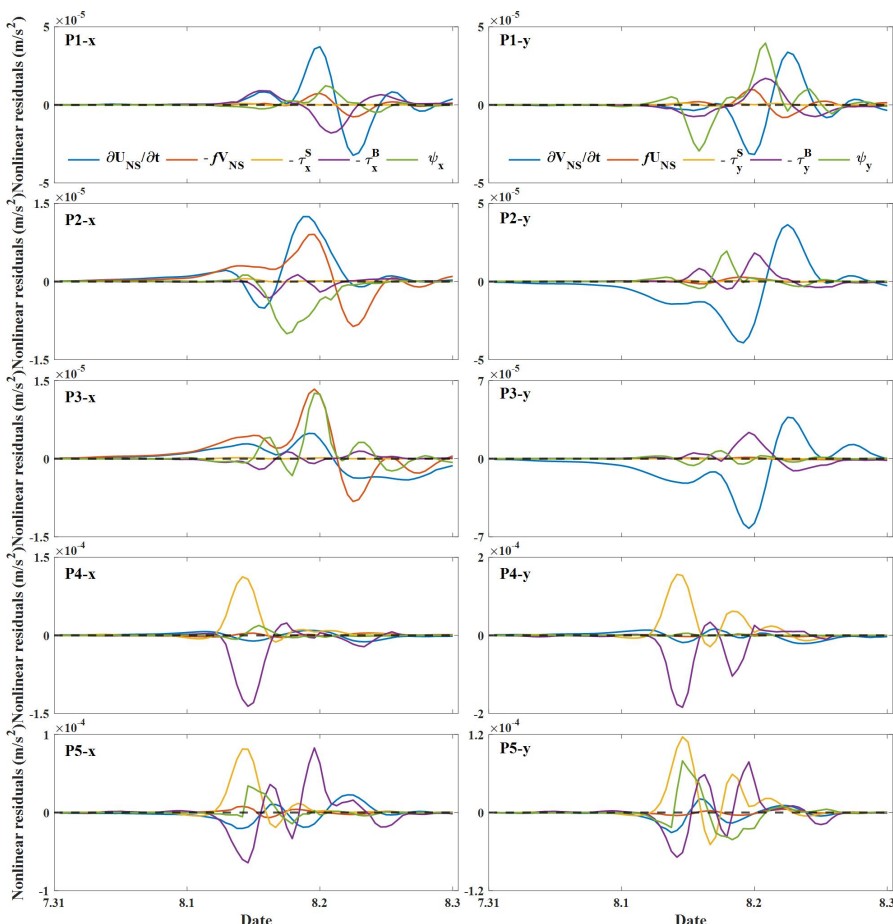

**Figure 6.** Time series of the nonlinear components of Typhoon Nida at P1, P2, P3, P4 and P5 in x direction (left) and y direction (right)

Given that P4 is located in the northern part of Qi'ao Island and P5 is located in Shenzhen Bay, these two points present areas of strong tide-surge nonlinear effect within the PRE region. As shown in Fig 7, in the eastward direction at P4, the nonlinear wind stress term plays a leading role and reached its positive maximum at 14:00 on 2 August 2016. The nonlinear bottom friction term is also significant. In the northward direction at P4, both the nonlinear wind stress and bottom friction terms contribute to the nonlinear effect. In the eastward direction at P5, the wind stress and bottom friction terms are more important than the other terms.

About all points, the results indicate that the gradient of the nonlinear level was stronger in the northerly direction than in the easterly direction (Hu et al., 2023), with the exception of the nonlinear Coriolis term. This study focuses on analyzing storm surges induced by Typhoon Nida using a two-dimensional model with high regulation unstructured grid, which enhances both accuracy and robustness of our conclusions. The establishment of direct mathematical relationships between nonlinear levels and dynamic terms through theoretical derivation provides



valuable insights. It was found that the nonlinear acceleration term mainly contributes to the top of
the bay, indicating a strong interaction between tidal current and storm-induced current (Song et
al., 2020). Additionally, the wind stress also affected the tide-surge interaction since the H was in
the denominator of wind stress term, especially in shallow water area, such as the northern part of
Qi'ao Island and Shenzhen Bay. Furthermore, it is noted that shallow water effects are more
significant in Shenzhen Bay because of limited water depth over tidal flats (Zheng et al., 2020).

## 3.4 Dynamic mechanism of the nonlinear residual levels influenced by different typhoon landfall time

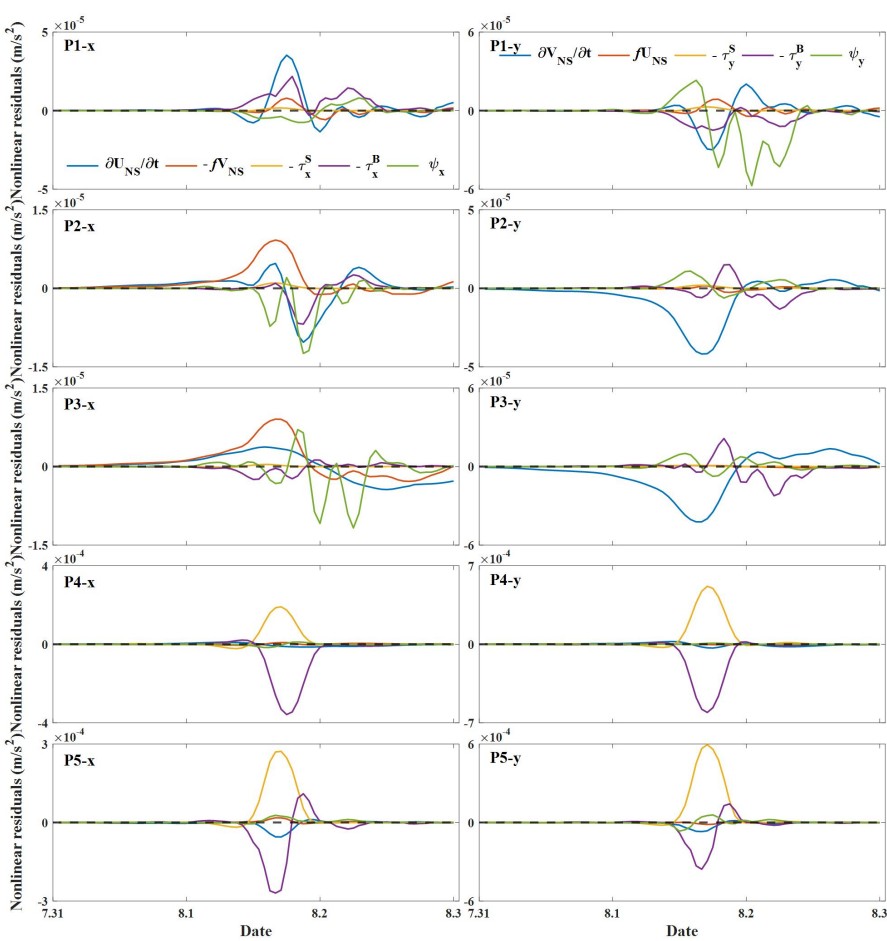

**Figure 7.** Time series of the nonlinear components at P1, P2, P3, P4 and P5 in x direction (left) and y direction (right) when maximum pure storm surge coincides with the HLW tidal phase

To investigate the nonlinear momentum characteristics resulting from different tidal forcing,
we calculated the nonlinear momentum terms. When the maximum increase in water level caused
by pure storm surge coincides with the HLW tidal phase, the temporal changes of nonlinear



residuals are shown in Fig 7. In the eastward direction at P1, the dominant factor is the nonlinear
local acceleration term, with some contribution from the nonlinear bottom friction term. The
effects of nonlinear advection and Coriolis force also make significant contributions. In the
northward direction, although the amplitude of the nonlinear local acceleration term remains large,
it is surpassed by the influence of the nonlinear advection term. Both in x-component and
y-component directions, wind stress terms exhibit weak impact and can be neglected. In the
eastward direction at P2, the values of various nonlinear terms were relatively small, contributing
little to the nonlinear effect, and the wind stress plays a minor role among all nonlinear terms. The
nonlinear Coriolis term contributes to positive nonlinear residuals while advection term and the
bottom friction term contribute negatively to nonlinear residuals. However, in the northward
direction, the local acceleration term plays a leading role and reaches its negative maximum in
nonlinear residuals at 16:00 on 1 August 2016. In the eastward direction at P3, the nonlinear
Coriolis and nonlinear advection terms make some contributions. However, in the northward
direction at P3, the dominant term is the nonlinear local acceleration term, with additional
contributions from bottom friction and nonlinear advection terms. The wind stress plays a minor
role among all nonlinear terms. In the eastward direction at P4, the nonlinear bottom friction term
plays a leading role, and reached its negative maximum at 18:00 on 1 August 2016. Following
closely behind is the nolinear wind stress term, which reached its positive maximum at 17:00 on 1
August 2016. In the northward direction at P4, both the nonlinear wind stress and bottom friction
terms contribute to the nonlinear effect. Specifically, while the bottom friction term make a
negative contribution to the nonlinear residuals, the wind stress contributes positively. The wind
stress term reached its positive maximum at 17:00 on 1 August 2016, and the bottom friction term
reached its negative maximum at the same time. In the eastward direction at P5, the wind stress
and bottom friction terms are more important than the other terms. In the northward direction at
P5, the wind stress term and bottom friction term also dominated the nonlinear residuals.
Moreover, the absolute value of wind stress term's positive maximum is greater than the absolute
value of bottom friction term's negative maximum. The advection term was so small that it was
recorded as negligible, resulting in a discontinuous time series.

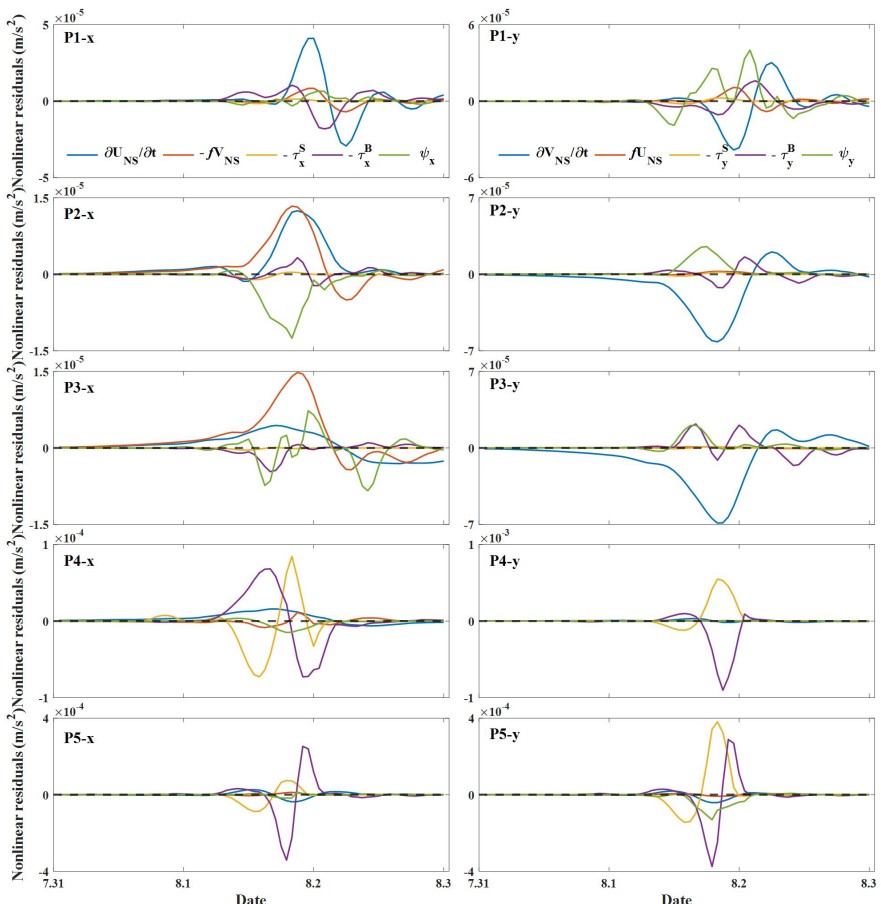

**Figure 8.** Time series of the nonlinear components at P1, P2, P3, P4 and P5 in x direction (left) and y direction (right) when the maximum pure storm surge coincides with the LHW tidal phase

When the maximum increase in water level caused by pure storm surge coincides with the LHW tidal phase, Figure 8 illustrates the time series changes of the nonlinear residuals. In the eastward direction at P1, the nonlinear local acceleration term also plays a leading role, followed by the nonlinear bottom friction term as the second-largest contribution. The effects of nonlinear advection and Coriolis force also make significant contributions. In the northward direction, the nonlinear local acceleration term and the nonlinear advection term make major contributions. Both in x-component and y-component, wind stress term are weak and can be negligible. In the eastward direction at P2, both bottom friction term and wind stress term exhibit significantly smaller compared to other terms. The positive nonlinear residuals are contributed by nonlinear local acceleration term while negative ones are result of contributions from nonlinear advection term and bottom friction term. However, in the northward direction at P2, it is observed that nonlinear local acceleration term plays a leading role and reached its negative peak value at 20:00 on 1 August 2016. In the eastward direction at P3, the nonlinear Coriolis term dominates the nonlinear residuals, while the nonlinear local acceleration term and the nonlinear advection term



also make some contributions. However, in the northward direction at P3, the nonlinear local acceleration term is predominant, with the bottom friction term and the nonlinear advection term also contributing. The wind stress and Coriolis effects are minimal. In the eastward direction at P4, the nonlinear bottom friction term and the nonlinear wind stress term have a greater impact than other terms. In the northward direction at P4, the nonlinear wind stress term and the bottom friction term significantly contribute to the nonlinear effect. Notably, the absolute value of the maximum of the bottom friction term is greater than that of the positive maximum of the wind stress term. The wind stress term reached its positive maximum at 20:00 on 1 August 2016, while the bottom friction term reached its negative maximum 1 h later. In the eastward direction at P5, the bottom friction term plays a leading role, with the wind stress term providing a secondary contribution. In the northward direction at P5, the wind stress term and the bottom friction term dominate the nonlinear residuals, as other terms have a negligible effect.

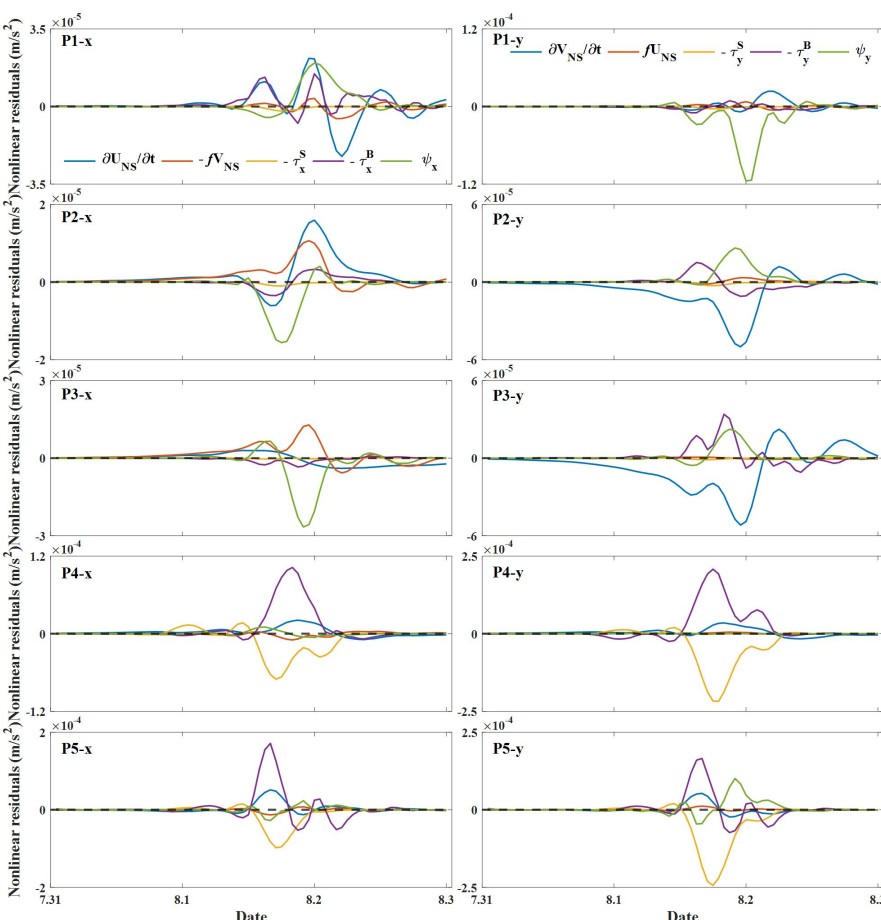

**Figure 9.** Time series of the nonlinear components at P1, P2, P3, P4 and P5 in x direction (left) and y direction (right) when the maximum pure storm surge coincides with the LLW tidal phase.





When the maximum increase in water level caused by the pure storm surge coincides with
the LLW tidal phase, the time series changes of nonlinear residuals are shown in Fig 9. In the
eastward direction at P1, the nonlinear local acceleration term, the advection term and the bottom
friction term make major contributions. In the northward direction, the nonlinear advection term
plays a leading role. Both in x-component and y component, wind stress term is weak and can be
negligible. In the eastward direction at P2, major contributions are made by the nonlinear local
acceleration term, Coriolis term, and nonlinear advection term. However, in the northerly direction
at P2, it is primarily influenced by the local acceleration term which reaches its negative peak in
nonlinear residuals at 23:00 on 1 August 2016. Meanwhile, the advection term makes positive
contrition to the nonlinear residuals, and reaches its positive maximum at 22:00 on 1 August 2016.
In the eastward direction at P3, the nonlinear advection term dominates, and the nonlinear Coriolis
term also makes some contributions. However, in the northward direction at P3, the nonlinear
local acceleration term is predominant, with the bottom friction and the nonlinear advection term
also contributing. The wind stress term and the Coriolis term are negligible in comparison. In the
eastward direction at P4, the nonlinear bottom friction term and the nonlinear wind stress term
make greater contributions than the other terms, with the nonlinear bottom friction term playing a
leading role and providing a positive contribution to the nonlinear residuals. In the northward
direction at P4, the nonlinear wind stress term and the bottom friction term significantly contribute
to the nonlinear effect, with the positive contribution of the bottom friction nearly balancing the
negative contribution of the wind stress. The wind stress reached its negative maximum at 19:00
on 1 August 2016, while the bottom friction term reached its positive maximum 1 h earlier. In the
eastward direction at P5, the bottom friction term plays a leading role, with the wind stress term
making the second-largest contribution. In the northward direction at P5, the wind stress term
makes a more significant contribution than the other terms.
Storm surges in the estuary area are influenced by typhoons and tides. The interaction
between these two forces on water levels has been demonstrated in many studies. The elevation of
storm surges in the PRE varies significantly depending on the tidal phase during which they occur.
Comparing nonlinear factors at those representative points across different tidal phases can make a
better understanding of the tide-surge interaction mechanism. Analysis of the nonlinear terms in
the residuals revealed that when storm surges coincide with high tides (HHW and LHW), the
nonlinear acceleration term predominantly affects the y-component at P1. Conversely, when storm
surges coincide with low tides (HLW and LLW) at P1, the nonlinear advection term plays a
leading role. In the eastward direction, the nonlinear acceleration term consistently plays a
dominant role at P1. In the northward direction, both P2 and P3 are characterized by a
predominant influence of the nonlinear acceleration term. When storm surges coincide with
different tidal phases, the contribution ratio of each nonlinear term remains almost unchanged, but
their magnitudes vary. The results illustrate that the primary source of the tide-surge interaction
nonlinear effects within the PRE is the effect of the tide's velocity. While in shallow water area,
such as the northern part of Qi'ao Island and Shenzhen Bay, the tide-surge interaction nonlinear
effects are predominantly influenced by a combination of wind and bottom friction.

## 4. Conclusions

An ADCIRC model has been utilized to simulate the storm surges in the PRE induced by



Typhoon Nida. Results from several numerical experiments investigating the interaction between
tides and storm surges indicate that when the tidal effect is incorporated, the simulations agree
well with observational data.
To study the characteristics of tide-surge interaction in the PRE, three types of model runs
were conducted, from which the total water level, the pure astronomical tide level, the pure storm
surge, the practical storm surge and the residual elevation due to the tide-surge interaction were
obtained. The results show that, the storm surge is significantly modulated by the tide level due to
the tide-surge interaction. A direct mathematical relationship between nonlinear levels and
dynamic influencing factors has been established. This derivation includes the local acceleration
term, the Coriolis force term, the wind stress term, the bottom friction term and the nonlinear
advection term. The nonlinear momentum term can reflect the momentum response of different
areas in the estuary to nonlinear effect. The momentum equation facilitates the establishment of a
relationship between the nonlinear factors of tide-surge interactions and the underlying physical
processes. By comparing nonlinear factors at representative points from the inner to the outer bay,
it has been shown that the local acceleration term and the nonlinear advection term predominantly
influence the nonlinear dynamics. However, in Shenzhen Bay and the northern part of Qi'ao
Island, the wind stress term and the bottom friction term emerge as the dominant nonlinear factors.
To further investigate the relationship between tide phases and storm surges, we adjusted the
landfall time of Typhoon Nida in our model simulations. The results shows that both the practical
storm surge level and the nonlinear level are significantly modulated by tidal forces, especially in
shallow water areas. The tidal contribution to the total water level surpasses the surge contribution
when the peak of pure storm surge coincides with high tides. Conversely, the contribution of tides
to the extreme total water level is less than the storm surge contribution when the maximum of
pure storm surge coincides with low tides. The analysis revealed that the nonlinear effect of
tide-surge interaction is positive when the peak of pure storm surge coincides with low tidal phase
(LLW and HLW). On the contrary, this nonlinear effect becomes negative when the pure storm
surge peak coincides with high tidal phase (HHW and LHW). Notably, when the maximum of
pure surges coincides with the LHW tidal phase, the contribution of the nonlinear effect is the
greatest compared to other tidal phases. The ratio of nonlinear residuals to the pure storm surge
and the ratio of the nonlinear residuals to the practical storm surge are both positive when the
maximum of the pure storm surge coincides with the HLW tidal phase, while the ratios during
other tidal phases are negative.
Although similar results were also presented in other studies (Song et al., 2020; Hu et al.,
2023), the discussions of nonlinear momentum for different tidal phases are still lacking. We
calculated the nonlinear terms to discuss the characteristics and mechanisms of tide-surge
interaction. When storm surges coincide with different tidal phases, the contribution ratio of each
nonlinear term remains almost unchanged, but their magnitudes are different. The results illustrate
that the primary source of the tide-surge interaction nonlinear effects within the PRE is the effect
of the tide's velocity. While in shallow water area, such as the northern part of Qi'ao Island and
Shenzhen Bay, the tide-surge interaction nonlinear effects are predominantly influenced by a
combination of wind and bottom friction.
Taking Typhoon Nida as a case study, the present research reveals the detailed characteristics
of tide-surge interaction in the PRE. The present results of this study can provide valuable
information for understanding the tide-surge interaction mechanism and improving storm surges



prediction within the PRE. However, further studies on additional typhoon events may be need, along with a comprehensive consideration of meteorological processes and the mechanisms of tidal-wave propagation within and outside the estuary, and the model system could still be improved in the future.

**Data availability.** The typhoon best track datasets used in this study is available from the CMA repository (http://tcdata.typhoon.org.cn). The tide gauge datasets used in this study are available from the authors on request.

**Author contributions.** LH, TZ, and SZ designed the study. LH conducted and made the analysis. All authors contributed to the discussion of the analysis and the final manuscript.

**Competing interests.** The contact author has declared that none of the authors has any competing interests.

**Acknowlegements.** We thank Hui Wang for him helpful discussions.

**Financial support.** This research was jointly funded by Independent research project of Southern Marine and Engineering Guangdong Laboratory (Zhuhai) (Grant No. SML2022SP301 and No. SML2022SP504); the National Natural Science Foundation of China (Grant No. 41976200, and 42206029); the Innovative Team Plan for Department of Education of Guangdong Province (No. 2023KCXTD015); the Guangdong Science and Technology Plan Project (Observation of Tropical marine environment in Yuexi), Guangdong Ocean University Scientific Research Program (Grant No. 060302032106).

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
