# Peer review of "Effect of nonlinear tide-surge interaction in the Pearl"

_EGUsphere, 2024_

## Author Comment (AC1)

Dear Reviewer:

We sincerely appreciate your time spent in providing constructive remarks and useful suggestions, which have significantly enhanced the quality of the manuscript and enabled us to make substantial improvements. Each suggested revision and comment from the reviewer has been carefully considered and incorporated. Below, you will find our point-by-point response to the reviewer's comments and revisions.

**1. The speed of the typhon's movement could also be investigated with this model, i.e. a slow-moving Nida. I miss also wind speed and wind direction.**

Response:
Thank you very much for your insightful comment and kind suggestion. In our actual typhoon simulations, we observed a significant increase in storm surge with even a small increase in forward speed for slow-moving typhoons (Park & Youn, 2021). As the storm moved faster, we noticed that the storm surges also increased (Park & Youn, 2021). While the moving speed of the typhoon center does affect the storm surges in the PRE, the effect is not significant (Du et al., 2020). Our focus in this paper is on the interaction between tides and storm surges, especially the tidal phases, rather than the factors inherent to typhoon itself.

Reference:
Park, Y. H., & Youn, D. (2021). Characteristics of storm surge based on the forward speed of the storm. Journal of Coastal Research, 114(SI), 71-75.
Du, M., Hou, Y., Hu, P., & Wang, K. (2020). Effects of typhoon paths on storm surge and coastal inundation in the Pearl River Estuary, China. Remote Sensing, 12(11), 1851.

**2. They mentioned, that the typhoon occurred during spring tide. Could the spring tide get higher, and the resulting total water level could be higher?**

Response:
Thank you very much for your helpful advice. The maximum storm surge induced by Typhoon Nida coincided with the highest high water (HHW), which represents the highest tidal level. As the tide gets higher, the total water level will also increases. The objective of this paper is to examine the tide-surge interaction under different tidal phases.

**3. I did not quite understand why the focus was only on the contributions from tide surge interaction.**

Response:
We sincerely apologize if our explanation caused any confusion. The interaction between tide and storm surge represents a pivotal element of storm surge dynamics, with a sophisticated underlying mechanism. In this paper, we aim to specifically focus on the interaction between tides and storm surges, as it clearly demonstrates the significance of this interaction. Through the research presented in this paper, it is also demonstrated that at P2, the contribution of the the nonlinear effect to the storm tide can reach up to 16.4%, and the contribution to the storm surge can reach -49.61%, which are important to the water levels.

**4. In the paper, I was sometimes confused about terminology. My impression was, that the**

**terminology storm surge included tides or not (line 20-21). My suggestion would be:**

**Total water level during this event, storm tide ($\zeta_{TS}$) = tide ($\zeta_T$) plus (atmospheric) surge ($\zeta_S$) plus tide-surge interaction elevation ($\zeta_I$)**

$$\zeta_{TS} = \zeta_T + \zeta_S + \zeta_I$$

**residual level, $\zeta_R$, the 'residual' = storm tide minus tide = (atmospheric) surge plus tide-surge interaction elevation**

$$\zeta_R = \zeta_{TS} - \zeta_T = \zeta_S + \zeta_I$$

**Well, I saw in other papers they used your terminology, but I still think this one is better.**

Response:

We sincerely apologize for any confusion and gratefully appreciate your valuable suggestion. In Line 20-21, we explain the three models used to distinguish the storm tide, storm surge, tide, and tide-surge interaction. The storm tide '$\zeta_{TS}$' in our paper is represented as '$\zeta_{T+S}$', which is simulated by both atmospheric and tidal forcing. The equation in Line 168-169 appears to be similar to yours. We have standardized the terminology in our manuscript. Specifically, we have replaced 'total elevation $\zeta_{T+S}$' with 'storm tide $\zeta_{TS}$', and 'tide-surge interaction $\zeta_I$' with 'nonlinear residual $\zeta_{Non}$'.

**5. Sometimes the authors changed the name of the variable in the paper. Please standardize the terms in the text and in the figures.**

Response:

Thank you for you kindness reminder, we have standardized the terminology in our manuscript as follows: 'total elevation $\zeta_{T+S}$' has been replaced with 'storm tide $\zeta_{TS}$', and 'tide-surge interaction $\zeta_I$' with 'nonlinear residual $\zeta_{Non}$'. Additionally, we have defined 'tide elevation' as $\zeta_T$, 'storm surge elevation' as $\zeta_S$, 'practical storm surge elevation' as $\zeta_{PS}$.

**6. I think with my above suggestions the abstract should be rewrite.**

Response:

We gratefully appreciate for your valuable suggestion. In this paper, we focus on tide-surge interactions and its mechanisms. we revised the abstract for better understanding. The new abstract is as follows:

'Storm surge is one of the most significant marine dynamic disasters affecting the coastal areas worldwide. A comprehensive study of its mechanisms is vital for improving forecasting capabilities and developing more effective prevention strategies. In this study, a two-dimensional (2D) numerical model based on the Advanced Circulation Model (ADCIRC) was employed to examine the characteristics of storm surges and the mechanisms of tide-surge interaction in the Pearl River Estuary (PRE) during Typhoon Nida (2016). Three distinct model runs were conducted to differentiate between variations in water levels attributable to astronomical tides, storm surges, and their combined effect. The results indicated that storm surges are primarily modulated by tides through tide-surge interactions. The nonlinear effect of tide-surge interaction is primarily generated by the nonlinear local acceleration term and convection term from the tide-surge interactions in the study area, as derived from the mathematical terms. However, in regions of shallow water, such as the northern part of Qi'ao Island and Shenzhen Bay, they are predominantly governed by the nonlinear wind stress term and bottom friction term. Furthermore, the variations in the y component of the nonlinear momentum terms are more significant than those in the x component. To investigate the impact of tidal phase on storm surge response to

Typhoon Nida, the timing of landfall was altered in order to introduce variations in PRE characteristics. The results demonstrate that the contribution ratio of each nonlinear term remains relatively constant, while the magnitudes exhibit fluctuations contingent on the timing of landfall. '

**7. Line:21 I think in this case you mean with the above definition storm tide**

Response:
We sincerely apologize for any confusion and thank you for your kind reminder. There may be some errors with the professional terminology, and we have taken the necessary steps to correct them in the manuscript.

**8. Line:36-39 Is this still so?**

Response:
Thank you for pointing out this question. In recent storm surge forecasts, many hydrodynamic models have considered the nonlinear tide-surge interaction, which is also the focus of this work. We describe them to emphasize the importance of tide-surge interaction.

**9. Line:59-66 In all cases the bathymetry and the local environment should be considered.**

Response:
Thank you for your valuable suggestion. It is widely recognized that the topography and local marine environment play an important role in influencing storm surges, a topic that has been extensively researched. In this paper, we aim to contribute to the existing research on the tide-surge interaction by investigating its contribution to the water levels.

**10. Line:67-68 This connection is not clear to me.**

Response:
We sincerely apologize for any confusion. The Pearl River Estuary (PRE), situated in Guangdong Province, South China, is believed to be one of the most extensive and significant estuaries in Asia, endowed with a distinctive geographical advantage and importance. The dynamics and tides of the PRE are greatly influenced by its intricate coastline, comprising numerous islands and other topographical features. This distinguishes the PRE from other estuaries that are directly discharged directly into the open shelf. Moreover, the PRE region is subject to typhoon activity on annual basis, which presents a significant risk to the regional economy and the safety of the population. The intricate topography gives rise to a more intricate interaction between tides and storm surges. It is both necessary and of great importance to investigate the interaction between tides and storm surges in this area.

**11. Line:77 I would say that good forecasting is needed to make forward-looking decision for coastal protection.**

Response:
We completely agree with you. We believe that good forecasting is indeed essential for making informed, forward-looking decisions regarding coastal protection. By predicting weather patterns, sea levels and potential storm surges, coastal managers and planners may be able to take proactive measures to safeguard communities and infrastructure against natural hazards.

**12. Line:80-81 How big is the difference between neap and spring tide at the Guangzhou gauge?**

Response:
Thank you for pointing out this question. At the Guangzhou gauge, there is approximately 3 m difference between neap and spring tides as shown in Fig.2. At the Chiwan gauge, the difference is approximately 2.8 m, and at the Hong Kong gauge, it is approximately 2.2 m between neap and spring tides.

**13. Line:98-111 The description of the Typhoon could be more detailed. Was the Typhoon a fast or slow-moving typhoon? I miss also wind speed and wind direction.**

Response:
Thanks for your kind advice, we have added some information about Typhoon Nida in the manuscript text as follows:
'As shown in Fig 1a, Typhoon Nida, classified as a sever tropical storm (STS) passed across the Philippines and entered the South China Sea (SCS) on July 31, 2016. It then proceeded westward and made landfall as a typhoon (TY) at 19:30 on August 1 in Shenzhen, Guangdong Province, China. The typhoon had a central pressure of 970 hPa and maximum wind speed exceeding 42 m/s. After that, it was weakening as a tropical storm (TS). On August 3 0:00, it as a tropical depression (TD) and dissipated.'
However, in this paper, we concerned the interaction between tide and storm surge, rather than delving into the intrinsic factors of the typhoon itself, such as its intensity, moving speed, landfall angle, and so on.

**14. Line:105 -106 This sentence belongs in the introduction.**

Response:
Thanks for your kind suggestion, we have incorporated information about Typhoon Nida into the introduction.

**15. Line:108 Figure 1a TD, TS, STS, TY?**

Response:
We apologize for our carelessness to illustrate the level of tropical cyclone (TC). According to China Meteorological Administration (CMA), the levels of TC are outlined in the following table. We have also included a new illustration in Fig 1 to clarify this further.

| Name | Wind speed (m/s) |
|---|---|
| TD (Tropical depression) | $10.8 \sim 17.1$ |
| TS (Tropical storm) | $17.2 \sim 24.4$ |
| STS (Sever Tropical storm) | $24.5 \sim 32.6$ |
| TY (Typhoon) | $32.7 \sim 41.4$ |
| STY (Sever Typhoon) | $41.5 \sim 50.9$ |
| Super TY (Super Typhoon) | $\geq 51.0$ |

[Figure]

**Fig 1** (a) The track and intensity of Typhoon Nida; (b) Model domain and grids of the study area; (c) Location and bathymetry of PRE. Stars represent the tidal gauges, and dots denote the calculation points of surge levels.

**16. Figure1c: The scale of the bathymetry is relatively smooth. Especially, the scale the bathymetry of the estuary could be better. Why do you have -10m ?**

Response:
We apologize for our carelessness that may have caused confusion. The bathymetric data has been sourced from nautical charts that have been made publicly available by the Hydrographic Department. We have revised the image, which is now depicted in Fig 1c.

**17. Line:158-162 How many model days does the model need to work?**

Response:
We conducted a 4-day simulation of this event, initiating with a cold start on July 30th. After a period of stabilisation lasting approximately half a day, we are pleased to present the results from 31 July to 3 August, as shown in Fig 3.

**18. Line:163-165 Is this important for the model setting?**

Response:
We apologize for that confused you. We intended to emphasize the importance of the tide at Lingding Bay, but it seems to be in a wrong place. We have moved this sentence to the introduction.

**19. Line:166-169 To make this sentence clearer, you should delete the explanation of 'practical storm surge elevation' and explain it in the next sentence. You should change the total water elevation $\zeta_{T+S}$ to $\zeta_{TS}$**

Response:

Thanks for your insightful comment and kind suggestion. We have replaced 'the total water elevation $\zeta_{T+S}$' to 'storm tide $\zeta_{TS}$', and standardized the names of the terms accordingly.

**20. Line:175-212 Actually, these are results.**

Response:

Thanks for your insightful comment and kind suggestion. We have placed this section in the 'Results' section of the paper.

**21. Line:184 Please, use the same y-axis for all stations, so from 0 to 5m**

Response:

We have redrafted this figure using the same y-axis as Fig 2 shows.

[Figure]

**Fig 2** Time series comparisons of measured and modeled astronomical tide levels at (a) Chiwan gauge

(b) Hongkong gauge (c) Guangzhou gauge

**22. Line:185 Is this really the modeled astronomical tide or the modeled total water level?**

Response:
Thank you for pointing out the issue. This is the modeled astronomical tide, during 7.2-7.30, in this period, there was no significant weather influence.

**23. Line:188-193 I am wondering that you compare observed water levels (including weather) with astronomical tides (without weather). The low water is not so good simulated. Why?**

Response:
Thank you for pointing out this question. The forecasting of storm surge has always been a critical issue. In recent years, various models have demonstrated their strengths and weaknesses in simulation. The ADCIRC model we have chosen performs well in simulating the maximum water level increase, although there are some inaccuracies before and after the peak water level. Nevertheless, this model is already highly developed and widely utilized in practice.

**24. Line:202-205 How do you define negative surge levels in this area? Please specify it clearer in the figure 3.   I have not understood this comment.**

Response:
We apologize for that confused you. We original intention was to convey the water level decrease caused by the storm. We have provide a revised description as follows.
'The numerical results show that the water level decrease is overestimated, resulting in significant errors in storm surge prediction. However, the simulated results for the positive extreme of storm tide closely match the observed values, demonstrating that the model used in this study effectively represents the tidal-surge interactions within the study area.'

**25. Line:207: The legends could be larger**

Response:
Thanks for your kindness advice. We have redrafted this figure with an enlarged legend and using the same y-axis as Fig 3 shows.

[Figure]

**Fig 3** Time series comparisons of measured and modeled storm surge levels at (a) Chiwan gauge (b) Hong Kong gauge (c) Guangzhou gauge

**26. Line:211 Table 1: What do you compare "tide plus pure storm surge" or the simulated total water level with the observations? Is there a difference between tide and storm surge or do mean the period of data?**

Response:
We apologize for that confused you. It's not 'tide plus pure storm surge', the 'storm surge' represents 'storm tide'. We are merely validating the model's simulation of tide and storm tide.

**27. Line:212 A table with the results for the three gauges and the five stations/points would be nice.**

Response:
Thank you for pointing out the problem. The three gauges contain observational data, while the five points are based on simulated data only. Consequently, we have selected the three gauges for the validation of our model.

**28. Line:215 The information, that the typhoon occurred during spring tide should be included in the motivation.**

Response:
Thanks for your kind suggestion. We also introduced it in the motivation. Here, we just want to emphasize the background again.

**29. Line:218 You should change storm surge to "total water level" or "storm tide"**

Response:
Thank you for your helpful suggestion. We have made the necessary revisions in the manuscript.

**30. Line:221 … as shown in Fig 1c… refers to the points 1-5 and not to the interaction.**

Response:
We apologize for that confused you. Here, we want to illustrate that these points represent its special area. P1 to P3 represent the internal, middle, and external of Lingding Bay, respectively. P4 represents the northern part of Qi'ao Island, and P5 represents the Shenzhen Bay, which are two shallow water area.

**31. Line:221 "Notably, the water depth at points P1, P2, and P3 exceeds 10 m, while the water depth at points P4 and P5 is less than 10 m." This information belongs in 2.4. Model setting. P1 to P3 look like a fairway channel. Are there changes if the points are 1 km away from the channel?**

Response:
Thank you for your kind suggestion. We selected these points to investigate the interaction between tides and storm surges, and we emphasize their water depth here for analyzing the results. Regarding the spatial characteristics of nonlinear residuals as shown in Fig 5, there appears to be little difference 1 km away from the channel.

**32. Line:224 You write about "nonlinear residuals levels" and in the figure 4b.,., the title is "nonlinear elevation" . Perhaps it is easier to use the term 'tide-surge interaction elevation' for both.**

Response:

Thank you for your helpful suggestion. We have standardized the names of the terms in the text.

**33. Line:227 Why is the decrease significant?**

Response:

Thank you for pointing out this question. Shenzhen Bay is a shallow water area. The effect of nonlinear tide-surge interaction can significantly modulate water levels in shallow regions. The shallow water effect arising from the nonlinear terms related to the total water depth in both the mass conservation equation and the momentum equations.

**34. Line:238 nonlinear residuals ($\zeta_{Non}$)   = nonlinear residual level ($\zeta_I$)**

Response:

Thank you very much for your suggestion. We have corrected all the terms unity accordingly.

**35. Line:242 maximum negative value = minmum?**

Response:

We apologize for that confused you. In fact, it does. Initially, we only wanted to emphasize the extremes values, which possess both negative and positive aspects. We have standardized the terminology throughout the text.

**36. Line:246 & 250 negative maxima = minimum?**

Response:

We are sorry for that confused you. In fact, it does. Initially, we only wanted to emphasize the extremes values, which possess both negative and positive aspects. We have standardized the terminology throughout the text.

**37. Line:214-281 I am wondering that the bathymetry, the wind direction and the wind speed are not included in the result chapter. I think they are very important for the calculation of the tide-surge interaction elevation. Due to the shallow bathymetry and the tides, there is an overestimation and underestimation of the surge. You can already see this in Figure 4. I miss also the regional aspects, e.g. whether the P4 or P5 are upwind or downwind of the wind direction. Also, it would be easier to understand the steps, if there some vertical lines in the time series marking the date of the 2D-images on the left side in Figure 4. The same y-axis could also help to compare the results (in all figures)**

Response:

Thank you for your insightful comments and valuable suggestion. The bathymetry of these five points are as follows: 15.26 m at P1, 16.73 m at P2, 22.50 m at P3, 1.67 m at P4, and 2.53 m at P5. The wind direction and the wind speed are shown in Fig 4. The wind speeds at these five points show little difference, which is likely related to the water depth. We have redraft the figure with same y-axis as shown in Fig 5.

[Figure]

**Fig 4** Wind velocity and wind direction at P1, P2, P3, P4, and P5

[Figure]

**Fig 5** Total water elevation (left) and nonlinear water elevation (middle) at different tidal phase; Time series of water levels for P1, P2, P3, P4 and P5 locations (right); The dashed line indicates the time corresponding to the time on the left and middle graphs.

**38. How do you define significant?**

Response:

We apologize for that confused you. We considered the extremes, the ratio of contributions, and the order of magnitude to define 'significant'. For example, the spatial nonlinear effect elevation was significant in Shenzhen Bay and the north part of Qi'ao Island as shown in Fig 5h. The nonlinear effect at P2 is significant, being the biggest value among the five points as shown in Table 2.

**39. Line:238 & 169 Is nonlinear residuals ($\zeta_{Non}$) = nonlinear residual level ($\zeta_I$) ?**

Response:

We apologize for that confused you. $\zeta_{Non} = \zeta_{I,}$ we have standardized the names of the terms in the revised manuscript accordingly.

**40. Line:278 Table 2 and Table 3**

**It would be nice to have the absolute maximum water levels.**

Response:

Thank you for your valuable suggestions. We have revised the titles of Tables 2-3, and corrected the name within the manuscript text.

**41. In table 2, for each point 1 to 5 you have calculated the contribution to the total high water.**

**The sum of tide and practical tide = 100% = total high water.**

**In the right column, we see the overestimation of surge, because**

**Surge + Nonlinear effect = practical surge.**

Response:

We apologize for that confused you. Well, we select the maximum value of the storm tide and calculate the contribution of surge, tide, practical surge, and nonlinear residual to the elevation of storm tide at the same time. C (storm surge, tide, practical surge, nonlinear residual) = elevation (storm surge, tide, practical surge, nonlinear residual) / elevation (storm tide). According to that, surge + tide + nonlinear effect = 100%. Absolutely, practical surge = storm tide - tide, so that surge + nonlinear effect =practical surge.

**42. From Figure 4 the total water level was lower than 2m at each point. The nonlinear effect is lower than 2m times 15 %= 0.3m. What do I learn for the forecast of storm tides?**

Response:

Thank you for your insightful comment. Although the nonlinear effect is smaller than the contribution caused by the storm, it should not be ignored. If we don't consider nonlinear effect between tide and storm surge, it could be increase the error in storm tide forecasting. Moreover, the mechanism of tide-surge interaction is complicated, which motivates us to do this work.

**43. Line:280 For what do I need table 3? What do I learn?**

Response:

Thanks for pointing out these problems. We emphasize the importance of the nonlinear effect, the ratio of the nonlinear residuals to the pure storm surge represents the extent to which the nonlinear effect amplifies or diminishes the direct impact of the storm surge.

**44. At least 75 % of the total water level is from the tide. Is table 3 important for the coastal protection?**

Response:

Sure, it is important. Table 3 reflects the contribution of the nonlinear effect to the storm surge. In the case of Typhoon Nida, the maximum storm surge coincided with the HHW tidal phase, which is a dangerous signal for storm tide. The contribution of the nonlinear effect to the storm surge is negative, which means it decreases the water level induced by the storm surge.

**45. Line:318 Table 6, I am wondering what is the basis of the calculation. How do you calculate 193.17 % practical surge for P1**

Response:

We apologize for that confused you. Well, we select the maximum value of the storm surge and calculate the contributions of the practical surge and nonlinear residual to the elevation of storm tide at the same time of different tidal phases. C ( practical surge, nonlinear residual) = elevation ( practical surge, nonlinear residual) / elevation (storm tide).

**46. Line:282-514. The authors did a lot of investigations, but for me it is not clear for what. How can you improve the prediction of water levels with your investigations. Haven't such studies been carried out for other coasts and estuaries?**

Response:

We apologize for that confused you. Our focus is on the tide-surge interaction, and we aim to identify the underlying mechanisms. The nonlinear effects of tide-surge interaction are important and relevant to the tide. Therefore, we adjusted the landfall time of Typhoon Nida to make the maximum storm surge coincide with various tidal phases.

**47. For me it would be more interesting to know how high the contribution to the total water level is and what influence the bathymetry has. It is described, but very briefly and more as a by-product.**

Response:

Thank you for your insightful comment and kind suggestion. Bathymetry is important for storm surge simulations, and we also consider it in our future work. This study focuses on the interaction between tides and storm surges under various tidal phases and explores the underlying mechanisms.

**48. What is new and help to improve the forecast models?**

Response:

Thank you for your insightful comment. We can improve the grid resolution, the precision of coastline, simulated parameter settings, and so on. At the same time, understanding the mechanism of tide-surge interaction also plays a certain role in storm surge forecasting.

---

## Author Comment (AC2)

Dear Reviewer:

We sincerely thank you for your time spent making your constructive remarks and useful suggestions, which has significantly enhanced the quality of the manuscript and enabled us to improve the manuscript. We carefully considered and incorporated each suggestion and comment from the reviewer. Please find below our point-by-point responses to the reviewer's comments as well as indications of the revisions made.

**General comments**

**In the presented study, the authors used a storm surge model to analyse the characteristics of Typhoon Nida in the Pear River Estuary (PRE). In order to gain a thorough understanding of how tidal forces, storm surges, and their nonlinear interactions influence total water levels in the region, the authors defined a series of scenarios (tidal forcing only, atmospheric forcing only, combined tidal and atmospheric forcing, and varying landfall). The results demonstrated that nonlinear tide surge-interactions were most significant when landfall times coincided with lowest high water (LHW). Moreover, the authors explored the various mechanisms underlying these nonlinear tide-surge interactions, identifying the local acceleration term and the nonlinear convection term as primary contributors, although other terms might dominate in shallow water areas.**
**While the title aptly reflects the paper's content and the study is compelling, there are several issues that should be addressed by the authors. Specific comments and suggestions are outlined below:**

**Specific comments**

**1. In my view, the manuscript includes an excessive amount of (sub)figures and tables, which detracts from its focus. The descriptions of all these (sub)figures and tables in the results section make that section challenging to read. It would enhance the paper if some of these figures and tables were moved to a Supplementary Material, thereby allowing a more concentrated and generalized description and discussion of the remaining key figures and tables. For instance, are Tables 4-7 all essential to the main manuscript? Additionally, is it necessary to present results for all five locations (P1-P5)? Given that some locations exhibit similar behaviour, focusing on two or three representative locations might suffice. This would streamline the manuscript and make the key findings more accessible to readers.**

Response:
Thank you for your insightful comments and kind suggestions. Tables 4 - 7 are necessary to the study. We would like to suggest that it might be beneficial to consider the nonlinear effect at different tidal phases. The ratio of the nonlinear residuals to the storm surge could be a useful way to represent the extent to which the nonlinear effect amplifies or diminishes the direct impact of the storm surge. These five points represent specific area. P1 to P3 represent the internal, middle, and external of Lingding Bay, respectively, and P4 representing the northern part of Qi'ao Island, and P5 representing Shenzhen Bay, which are two shallow water areas.

**2. As you mentioned, there are already comparable studies in similar settings or even specifically focusing on the Pear River Estuary (e.g., Hu et al., 2023). How does your study compare to these previous studies and what are the novel contributions of your research?**

**These aspects should be more clearly highlighted in the manuscript.**

Response:

Thank you for your insightful comments and helpful suggestions. Our study is primarily concerned with the interaction between tides and storm surges, with a particular focus on the influence of tidal phases on the nonlinear effects. Additionally, our research offers a refinement of the momentum terms by distinguishing between wind induced friction and bottom induced friction, which represents an advancement over the work of Hu et al. (2023) and Yang et al. (2019). The results show that the wind stress term and bottom friction term played different role in the process.

Reference

Yang, W., Yin, B., Feng, X., Yang, D., Gao, G., and Chen, H.: The effect of nonlinear factors on tide-surge interaction: A case study of Typhoon Rammasun in Tieshan Bay, China. Estuar. Coast Shelf S, 219, 420-428, https://doi.org/10.1016/j.ecss.2019.01.024, 2019.

Hu, S., Liu, B., Hu, M., Yu, X., Deng, Z., Zeng, H., and Li, D.: Quantification of the nonlinear interaction among the tide, surge and river in Pearl River Estuary, Estuar. Coast Shelf S., 290, 108415, https://doi.org/10.1016/j.ecss.2023.108415, 2023.

**3. An outlook of your results would be beneficial. Are there any findings that can be generalized and applied to other regions? Or how do your results contribute to improving forecasting skills, as mentioned in your abstract? Providing potential applications of findings will only increase the impact of a study!**

Response:

Thank you for your insightful comment and kind suggestion. The PRE is characterized by its distinctive funnel-shaped bay, where the dynamics and tides are greatly influenced by its complex coastline, which is composed of numerous islands and other features, setting it apart from estuaries that are discharged directly into the open shelf. Our results are also applicable to similar geographical environments and typhoon tracks. Although our work is based on a specific area, it has been found in the research of others that nonlinear effects are strongest at top of the bay (Yang et al., 2019). In our experiments, we took into account the number of tidal harmonic constituents, as well as the rendering of the topography, in order to enhance the accuracy of the simulation. This model has also been applied to storm surge forecasting, with satisfactory simulation results.

Reference

Yang, W., Yin, B., Feng, X., Yang, D., Gao, G., and Chen, H.: The effect of nonlinear factors on tide-surge interaction: A case study of Typhoon Rammasun in Tieshan Bay, China. Estuar. Coast Shelf S, 219, 420-428, https://doi.org/10.1016/j.ecss.2019.01.024, 2019.

**4. Please elaborate on why Typhoon Nida was chosen for your study. Would the results differ significantly for typhoons with other tracks and intensities? Do your findings represent general characteristics of the PRE during storm surges, or are they only specific to events similar to Typhoon Nida? This should be discussed in detail.**

Response:

Thank you for your insightful comments. We selected Typhoon Nida for our study because the maximum storm surge induced by Nida coincided with the highest high water (HHW). Additionally, we conducted simulations for Typhoon Hato and Typhoon Mangkhut to investigate the underlying the mechanisms. As shown in Figs 1-2, in the momentum analysis for these events, similar to Nida, suggests that the nonlinear effect is mainly generated by the nonlinear local acceleration term and the convection term resulting from the tide-surge interactions in the study area. Additionally, it seems that variations in the y component of the nonlinear momentum terms are more significant than those in the x component, which is similar to what was analysis conducted through Nida. So, these findings reflect the general characteristics of the PRE during storm surges. Meanwhile, our other study delves into the specifics of Typhoon Hato and Typhoon Mangkhut.

[Figure]

**Fig 1** Time series of the nonlinear components of Typhoon Hato at P1, P2 and P3 in x direction (left) and y direction (right)

[Figure]

**Fig 2** Time series of the nonlinear components of Typhoon Mangkhut at P1, P2 and P3 in x direction (left) and y direction (right)

**5. In the description of the numerical model, there is no mention of whether a 2D or 3D model was applied. This information is only found towards the end of the paper. Please include this detail in the initial description of the numerical model. Additionally, please briefly explain why a 2D model was deemed sufficient for your model domain?**

Response:

Thank you for your insightful comments and helpful suggestions. We opted to use a two-dimensional (2D) hydrodynamic ADCIRC model. The vertical current shear in well-mixed environment at shallow water depth is relatively small. This suggests that a 2D depth averaged model is sufficient to revel the physical processes of tide-surge interaction (Idier et al., 2012; Song et al., 2020; Zhang et al., 2017). We have introduced them in section 2.3, where you will find further details on this topic.

Reference

Idier, D., Dumas, F., and Muller, H.: Tide-surge interaction in the English Channel, Natural Hazards Earth System Sciences, 12, 3709-3718, 2012.

Song, H., Kuang, C., Gu, J., Zou, Q., Liang, H., Sun, X., and Ma, Z.: Nonlinear tide-surge-wave interaction at a shallow coast with large scale sequential harbor constructions, Estuarine, Coastal Shelf Science, 233, 106543, 2020.

Zhang, H., Cheng, W., Qiu, X., Feng, X., and Gong, W.: Tide-surge interaction along the east coast of the Leizhou Peninsula, South China Sea, Continental Shelf Research, 142, 32-49, 2017.

**6. Is it truly necessary to include all these formulas? When discussing a numerical model like ADCIRC, providing a reference for readers to find additional implementation details should be sufficient. Additionally, common metrics such as the RMSE are generally**

well-known to readers, so including their formulas is redundant. It would be more effective to only focus on the formulas that are essential for understanding your specific work (e.g., Formula 12).

Response:

Thank you for your insightful comment and kind suggestion. While many readers are familiar with metrics such as the root mean square error (RMSE) and the correlation coefficient (R), it is possible that not everyone is as acquainted with Skill. Therefore, we have taken the decision to delete the formulas for RMSE and R, but to retain the formula for Skill in order to ensure clarity for all readers.

**7. I also have difficulty understanding some of your terminology. Could you clarify what is meant by terms such as "negative/positive surge levels" and "negative maximum"?**

Response:

We apologize for that caused any confusion. Our initial intention was to emphasize the extreme values resulting from both increased and decreased water levels, which cloud have either negative or positive implications. We have amended the manuscript to address this issue as follows:
'The numerical results shows that when Nida approached to the PRE, the simulation of increased water levels was underestimated, resulting in significant errors in prediction storm tides. However, the simulated results for maximum water levels closely match the observed values, demonstrating that the model employed in this study effectively represents the tidal-surge interactions within the study area.'

**8. LL82-84: "In this paper, we utilize a recently developed ADCIRC based PRE surge model, which is nested within the China Sea tide and surge model, to investigate the mechanism of tide-surge interaction."**
**Could you specify which China Sea tide and surge model is being referred to here? If there is an existing publication and a reference for this model, please include it here.**

Response:

We apologize for any confusion this may have caused. This model is independent design based on the ADCIRC model and represents a distinctive approach to simulating storm surges in PRE. It takes into account both tides and storm surges, thus offering a more comprehensive representation of this phenomena.

**Technical corrections**

**1. The manuscript would greatly benefit from some language editing. Below, you will find an incomplete list of issues that I have noticed. One recurring issue, for instance, is the inconsistent use of articles. Here are a few examples:**
**"Advanced Circulation Model […]" should be "The Advanced Circulation Model […]**
**"[…] while advection term […]" should be "[…] while the advection term […]"**
**"[…] makes positive contribution […]" should be "[…] makes a positive contribution […]"**

Response:

Thank you for your constructive comment and kind reminder. We have reviewed the manuscript and made the necessary corrections.

**2. LL60-62: "Rego and Li (2010) studied the storm surge induced by Hurricane Rita revealed that the advection terms were dominant over bottom friction with significant spatial-temporal variations in the nonlinear terms."**
**"[…] Hurricane Rita and revealed that […]"**

Response:

Thank you for your kind reminder. We have checked the manuscript and corrected the mistakes.

**3. LL67-68: "The characteristics of storm surges and nonlinear effects in the Pearl River Estuary (PRE) are especially complex, as the PRE is one of the most important economic regions of China."**
**The first and second parts of this sentence are not logically connected. In my opinion, it would be better to connect this sentence with the following one: "The characteristics of storm surges and nonlinear effects in the Pearl River Estuary (PRE) are especially complex, as its topography consists of deep channels, shallow shoals, and tidal flats […]."**

Response:

Thank you for your kind suggestion, we appreciate it gratefully, and have corrected the mistakes in our manuscript.

**4. L98: "2.1 Typhoon NIDA"**
**Why are capital letters used for Typhoon Nida here?**

Response:

We are apologize for any confusion caused by our careless mistake. We have checked the manuscript and corrected the mistake.

**5. LL99-100: "Typhoon Nida generated in the western North Pacific Ocean on 29 July 2016 and began to move westward rapidly."**
**"Typhoon Nida was generated […]"**

Response:

Thank you for your kind suggestion. We appreciate it gratefully, and have made the necessary corrections to the text.

**6. Figure 1 Please consider that some readers may have colour vision deficiencies. Therefore, it is advisable to avoid using 'jet' colourmaps in your figures. Additionally, you should ensure that all figures are checked for appropriate colour choices and contrast. It is also important to define all abbreviations used in the figures, such as TD, TS, STS, and TY.**

Response:

Thank you for your kind suggestions. We have revised this figure and have added an explanation of the typhoon levels, as shown in Fig 3. We have also added some information about Typhoon Nida to the manuscript text as follows:

'As shown in Fig 1a, Typhoon Nida, classified as a sever tropical storm (STS) passed across the Philippines and entered the South China Sea (SCS) on July 31, 2016. It then proceeded westward and made landfall as a typhoon (TY) at 19:30 on August 1 in Shenzhen, Guangdong Province, China. The typhoon had a central pressure of 970 hPa and maximum wind speed exceeding 42 m/s. After that, it was weakening as a tropical storm (TS). On August 3 0:00, it as a tropical depression (TD) and dissipated.'

[Figure]

Fig 3 (a) The track and intensity of Typhoon Nida; (b) Model domain and grids of the study area; (c) Location and bathymetry of PRE. Stars represent the tidal gauges, and dots denote the calculation points of surge levels.

**7. LL115-116: "Which is unstructured triangular grids in the horizontal plane to resolve dynamics in complex shorelines."**
**This sentence is unclear. A better way to phrase it might be: "Unstructured triangular grids were used in the horizontal plane […]"**

Response:

Thank you for your kind suggestion, and we appreciate it gratefully. We have rephrased the sentence following your suggestions.

**8. LL134: "2.3 wind field of typhoon"**
**Capitalisation should be used at the beginning of your header. However, I believe that introducing a new subchapter for the wind field model may be unnecessary, as this information would seamlessly fit into the previous subchapter.**

Response:

Thank you for your insightful comment and useful suggestion. We have taken your suggestion and have made the necessary revisions accordingly.

**9. LL135-136 "We employed the analytical wind model from Holland (1980), which has**

**applied in reconstructing the wind field during Typhoon Nida."**
**"[…] which was applied for reconstructing […]"**

Response:

Thank you for your kind suggestion. We appreciate it gratefully, and have made the necessary corrections to the manuscript.

**10. LL163-164: "As a semi-enclosed bay, Lingdingyang Bay is regularly affected by both storm surges and irregular semi-diurnal tides."**
**Not everyone is familiar with your study area. Ideally, all relevant geographical names should be shown on a map of your study area.**

Response:

Thank you very much for the positive feedback and constructive suggestions. We have redrafted this figure and added the relevant geographical names as shown in Fig 3.

**11. Figures 2 and 3 What do the lines and points represent in Figures 2 and 3? It is not explained, which elements correspond to the measurements and which to the simulations.**

Response:

We apologize for that confused you, and thank you for your kind reminder. The points in Figure 2 represent the observed values, while the lines represent the simulated values. Figure 3 (manuscript) also represents observed values with points and simulated values with lines, in a similar to Figure 2 (manuscript). We have redrafted these figures as shown in Figs 4-5.

[Figure]

**Fig 4** Time series comparisons of measured and modeled astronomical tide levels at (a) Chiwan gauge (b) Hongkong gauge (c) Guangzhou gauge

[Figure]

**Fig 5** Time series comparisons of measured and modeled storm surge levels at (a) Chiwan gauge (b) Hong Kong gauge (c) Guangzhou gauge

**12. LL227-228: "At the same time, the nonlinear residual levels shows that it is negative in Lingdingyang Bay, except for its top region (Fig 4e)."**
**"[…] the nonlinear residual levels show that […]"**

Response:

Thank you for your kind suggestion. We appreciate it gratefully, and have corrected the mistakes in the manuscript.

**13. LL240-243: "When the typhoon landfall, the nonlinear residual levels peaked at their maximum positive value and subsequently reached their maximum negative value before the water level experienced its most substantial increase."**
**"When the typhoon made landfall, […]"**

Response:

Thank you for your kind suggestion. We appreciate it gratefully, and have corrected the mistakes in the text.

**14. Figure 4 Highlighting the times shown in the left and middle panels within your right panels would enhance clarity.**

Response:

Thank you for your insightful comment and helpful suggestion. We have redrafted this figure, including the addition of dashed lines, as shown in Fig 6.

[Figure]

**Fig 6** Total water elevation (left) and nonlinear water elevation (middle) at different tidal phase; Time series of water levels for P1, P2, P3, P4 and P5 locations (right); The dashed line indicates the time corresponding to the time on the left and middle graphs.

**15. Table 6 Could you clarify what the percentages in Table 6 actually represent? Are these percentages indicative of changes compared to your baseline scenario?**

Response:

We apologize for any confusion caused. The contributions of the practical storm surge and the nonlinear effect to the total elevation (we would use 'storm tide' replace it) are analyzed with different landfall times. This analysis aims to gain a deeper understanding of the role played by tide-surge interaction, taking into account the influence of different tidal phases.

**16. LL367-369: "In the eastward direction at P2, the values of various nonlinear terms were relatively small, contributing little to the overall nonlinear effect, with the wind stress term plays a minor role of all nonlinear terms."**
**"[…] with the wind stress term playing a minor role among all nonlinear terms."**

Response:

Thank you for your kind suggestion. We appreciate it gratefully, and have corrected the mistakes in the manuscript.

**17. LL372-373: "In the eastward direction at P3, the nonlinear Coriolis dominated, the values reached its positive maximum at 23:00 on 1 August 2016."**
**"[…] the nonlinear Coriolis term dominated, with values reaching […]"**

Response:

Thank you for your kind suggestion. We appreciate it gratefully, and have corrected the mistakes in the manuscript.

**18. LL449-451: "In the eastward direction at P2, both bottom friction term and wind stress term exhibit significantly smaller compared to other terms."**
**"[…] wind stress term are significantly smaller […]"**

Response:

Thank you for your helpful suggestion. We appreciate it gratefully, and have corrected the mistakes in the manuscript.

**19. LL561-564: "However, further studies on additional typhoon events may be need, along with a comprehensive consideration of meteorological processes and the mechanisms of tidal-wave propagation within and outside the estuary, and the model system could still be improved in the future."**
**"[…] events may be needed […]"**

Response:

Thank you for your kind suggestion. We appreciate it gratefully and have corrected the mistakes in the manuscript.

---

## Author Response (AR2)

Dear Editor:

We are truly grateful to you and the reviewers for your meticulous and comprehensive examination of our manuscript, 'Effect of nonlinear tide-surge interaction in the Pearl River Estuary during Typhoon Nida (2016)', MS No. egusphere-2024-1940. We sincerely appreciate the reviewers providing constructive remarks and useful suggestions, which have significantly enhanced the quality of the manuscript and enabled us to make substantial improvements. Each suggested revision and comment from the reviewer has been carefully considered and incorporated. Please find below our point-by-point response to the reviewer's comments and revisions.

**Comments for Reviewer 1:**

**1. In the paper I miss the time series of wind direction and wind speed for the trumpet-shaped bay. The authors investigate wind surge, which depends on the wind direction in this area. It is important to know, when there was onshore or offshore wind.**

Response:

Thank you for your insightful comment and kind suggestion. We fully agree that understanding the temporal variations in wind direction and speed is crucial for accurately investigating wind surges, as these phenomena are significantly influenced by the characteristics of the wind field within the region. Specifically, onshore winds tend to enhance storm surges, while offshore winds can mitigate their impact. Unfortunately, due to the specific geographic and topographic features of the trumpet-shaped bay, comprehensive time series data for wind direction and speed may not be readily available in the public domain. However, the reconstruction of the wind field from Holland (1980) itself is based on observational or reanalysis data, which has a relatively high credibility (Yang et al., 2019; Hu et al., 2023; Sebastian et al., 2024).

The model wind has been added in Figure 4 of the manuscript, and the spatial and temporal characteristics of the storm tide has been described (Line 229-246), as follows.

'At 9:00 on 1 August, during the lowest low water (LLW) tidal phase, the PRE area showed a decrease in water level which was influenced by the tide (Fig 4a). The offshore wind exhibited a tendency to decrease the water level. While the nonlinear residuals were positive in Shenzhen Bay and the northern part of Qi'ao Island (Fig 4b). At 15:00 on 1 August, which coincides with the lowest high water (LHW) tidal phase, the total water elevation in PRE exhibited a negative to positive trend from northeast to southwest, the offshore wind make some contribution. The most notable decline in water level is observed in Shenzhen Bay (Fig 4d). At the same time, the nonlinear residuals are negative throughout Lingding Bay, with the exception of its upper region (Fig 4e). At 19:00 on 1 August, during the highest low water (HLW) tidal phase, both offshore wind and tide conditions resulted in a negative trend in the elevation of storm

tide in the PRE area, with the most significant negative values occurring from northeast to southwest. Notably, the most substantial decrease in water level was observed in Shenzhen Bay (Fig 4g). While the nonlinear residuals are positive, their impact is particularly significant in Shenzhen Bay and the northern part of Qi'ao Island (Fig 4h). During the HHW tidal phase, the storm tide elevation in the PRE area exhibits a most substantial increase (Fig 4j), and the onshore wind enhance storm surges (Fig 4l). Conversely, during the same phase, the nonlinear residuals exhibit a most significant decrease (Fig 4k). Furthermore, at 10:00 on 2 August, during the LLW tidal phase, the storm tide elevation in the PRE area was negative which was predominantly attributed to the tide. The onshore wind exhibited a negligible effect on the water level, while the nonlinear resudials were positive.'

Reference

Hu, S., Liu, B., Hu, M., Yu, X., Deng, Z., Zeng, H., and Li, D.: Quantification of the nonlinear interaction among the tide, surge and river in Pearl River Estuary, Estuar. Coast Shelf S., 290, 108415, https://doi.org/10.1016/j.ecss.2023.108415, 2023.

Sebastian, M., Behera, M. R., Prakash, K. R., and Murty, P.L.N.: Performance of various wind models for storm surge and wave prediction in the Bay of Bengal: A case study of Cyclone Hudhud. Ocean Engineering, 2024, 297: 117113, https://doi.org/10.1016/j.oceaneng.2024.117113.

Yang, W., Yin, B., Feng, X., Yang, D., Gao, G., and Chen, H.: The effect of nonlinear factors on tide-surge interaction: A case study of Typhoon Rammasun in Tieshan Bay, China. Estuar. Coast Shelf S, 219, 420-428, https://doi.org/10.1016/j.ecss.2019.01.024, 2019.

[Figure]

**Figure 4.** Storm tide elevation and wind vector (left), nonlinear level (middle) at different tidal phase and time series of water elevations (right) for P1, P2, P3, P4 and P5 locations; The dashed line indicates the time corresponding to the time on the left and middle graphs.

**2. For example, in Figure 4, you see the tide is higher than the storm tide for 1 Aug, because of the offshore wind (P1, P2, P5). Please use an offshore station for the wind direction. In this complex area the wind direction is very important. Also, observations of wind direction should compared with modelled wind direction.**

Response:

Thank you for pointing out this question. We fully agree that accurate wind

direction data are crucial for understanding the complex interactions between offshore winds and storm surges in the study area. We acknowledge that obtaining direct offshore wind direction data from the study area presents significant challenges due to the lack of available observational stations. We have provided an explanation of the impact of wind and tide on storm surges as outlined in response to comment 1.

**3. Figure 4, please check the legends and titles.**

Response:

Thank you for your kind suggestion. We appreciate it gratefully and have duly made the necessary corrections, as shown in Fig 4. Specifically, we have replaced 'Nonlinear elevation' with 'Nonlinear level' in the title of Fig 4b.

**4. In my review (number 27) I suggested to see some results from the model. I'm still interested in the model data for table 2.**

Response:

We appreciate your interest in enhancing the robustness of our study. It should be noted that the primary focus of our research is the Lingding Bay area, so we selected five points to represent the special area with the intention of providing an illustration of the nonlinear effect at the internal, middle, external of the bay, and the shallow water area. The three gauge stations were used exclusively for the purpose of validating the water level between simulations and observations. The contributions of these stations have been calculated as illustrated in the following table. For Guangzhou station, the contribution of wind, tide and nonlinear effect are the most significant among the three stations.

**Table** The contribution to the storm tide elevation

| Stations | Storm surge (%) | Tide (%) | Practical storm surge (%) | Nonlinear effect (%) |
| --- | --- | --- | --- | --- |
| Chiwan | 32.57 | 85.3 | 14.7 | -17.88 |
| Hong Kong | 23.53 | 80.94 | 19.06 | -4.46 |
| Guangzhou | 40.33 | 88.18 | 11.82 | -28.51 |

**5. Line 260 – 268 I do not understand this paragraph. From my point of view, the surge (wind) is overestimated because of the flat bathymetry.**

Response:

We sincerely apologize if our explanation caused any confusion. In our research, we have carefully considered the influence of bathymetry on the storm surge modeling. The north of Qi'ao Island and the Shenzhen Bay are affected by the shallow water effect. The shallow water effects is expected to play an important role in the generation of the tide-surge interaction (Zhang et al., 2017). A division of the nonlinear terms from the momentum equation reveals that the tide-surge interaction nonlinear effects within the PRE are primarily caused by the velocity of the tide. In

areas of shallow water, such as the northern part of Qi'ao Island and Shenzhen Bay, the tide-surge interaction nonlinear effects are predominantly influenced by a combination of wind and bottom friction.

Reference

Zhang, H., Cheng, W., Qiu, X., Feng, X., and Gong, W.: Tide-surge interaction along the east coast of the Leizhou Peninsula, South China Sea, Continental Shelf Research, 142, 32-49, 2017.

**6. 3.3 Changing the time of landfall means a wind direction and wind speed change in PRE.**

Response:

We sincerely apologize for any confusion. We would like to clarify that in our study, changing the landfall time was specifically designed to align the maximum storm surge with different tidal phases, with the aim of investigating the interaction between storm surge and tidal conditions more effectively. Importantly, the wind speed and wind direction themselves were not altered by this change in landfall time. The primary effect was on the tidal phase during which the storm surge occurred, allowing us to explore the combined impact of storm surge and tidal conditions under various scenarios. It is means the nonlinear interaction has a significantly negative correlation with the tidal level (Zhang et al., 2021).

Reference

Zhang, X., Chu, D., and Zhang, J.: Effects of nonlinear terms and topography in a storm surge model along the southeastern coast of China: a case study of Typhoon Chan-hom, Natural Hazards, 107, 551-574, 2021.

**7. My question about this experiment would be, what would your experiment look like, if the track of the typhoon were more south-easterly, i.e. landfall 113°E and 22°N?**

Response:

Thank you for pointing out this question. We would like to clarify that in our study, the alteration of the landfall time was specifically designed to align the maximum storm surge with different tidal phases. This modification was made to investigate the interaction between storm surge and tidal conditions more effectively. It is noteworthy that the wind speed and wind direction were not altered by this adjustment of the landfall time. Instead, the primary effect was on the tidal phase during which the storm surge occurred, allowing us to explore the combined impact of storm surge and tidal conditions under various scenarios. This paper explores the nonlinear interactions between typhoons and tides in the Pearl River Estuary. It is important to note that if the landfall location of the typhoon is altered, the tidal state at the landfall point is also likely to change, which could potentially lead to an increase

in nonlinear factors and a more complex process. The impact of TC tracks, such as different landfall locations, approach directions and speeds, has also been considered in other work.

**8. I wonder how the analyses in chapters 3.2-3.4 can be carried out without wind direction and wind force. How are you going to convince me that you are drawing the right conclusions for the bay? In my opinion the surge is very dependent on the wind direction for this trumpet-shaped bay.**

Response:

Thank you for pointing out this question. The wind direction and wind speed were reconstructed using empirical formulas based on typhoon data from the China Meteorological Administration (CMA). This technique is a commonplace one in storm surge modeling, as outlined in detail in chapter 2.3 of the manuscript. The typhoon landfall time was modified to align the maximum storm surge with various tidal phases, while maintaining the wind speed and direction constant. Utilising mathematical formulas, a comparison was made between the nonlinear interactions between storm surge and astronomical tide under different phase combinations. It was determined that the landfall time exerts a significant influence on surge peak through tide-surge interactions (Zhang et al., 2022).

Reference

Zhang, Z., Guo, F., Song, Z., Chen, P., Liu, F., and Zhang, D.: A numerical study of storm surge behavior in and around Lingdingyang Bay, Pearl River Estuary, China. Natural Hazards, 2022: 1-26, https://doi.org/10.1007/s11069-021-05105-w

**Comments for Reviewer 3:**

**1. Lines 046–047: The present work focuses on the mechanisms and behaviors of water levels in estuary areas, which is a valuable contribution. However, when describing certain phenomena as "widely known," it's important to do so with caution. For instance, on beaches and barrier islands, the contribution of wave runup to total water levels is significant and cannot be overlooked. Studies such as Hsu et al. (2023; https://doi.org/10.5194/nhess-23-3895-2023) and Vicens-Miquel et al. (2025; https://doi.org/10.2112/JCOASTRES-D-24-00016.1) have highlighted the critical role of wave runup during extreme weather events. Including references to such relevant research in the introduction would enhance the context and provide a more comprehensive perspective.**

Response:

Thank you for your constructive comment and kind reminder. We have reviewed the manuscript and made the necessary corrections (Line 49). The revisions are as follows:

'The total water level can be divided into three main components ...'

We have added more references to illustrate the current state of research on storm

surges (Line 45-48). The revisions are as follows:

'During periods of extreme water levels, nonlinear interactions occur within the estuarine area among tide, storm surge, wave, and river streamflow (Hu et al., 2023). For instance, on beaches and barrier islands, the contribution of wave runup to total water levels is significant and cannot be overlooked (Vicens-Miquel et al., 2025).'

**2. Lines 106–107: Did the authors intend to refer to a "severe tropical storm (STS)"? Additionally, the statement could be clarified by revising it as follows: "As shown in Fig. 1a, Typhoon Nida, classified as a severe tropical storm (STS), passed over the Philippines and entered the South China Sea (SCS) on July 31, 2016."**

Response:

Thank you for your kind suggestion. We appreciate it gratefully, and have made the necessary corrections to the manuscript (Line 106-108). The revisions are as follows:

'As shown in Fig 1a, Typhoon Nida classified as a sever tropical storm (STS), passed across the Philippines and entered the South China Sea (SCS) on July 31, 2016.'

**3. Lines 139–155: The analytical wind and pressure model proposed by Holland (1980) has been widely utilized in various experiments. Later, Holland (2008; https://doi.org/10.1175/2008MWR2395.1) introduced an updated analytical model that incorporates the effects of storm translation speed on hurricane wind and pressure. While studies have demonstrated that the revised model provides improved accuracy in accounting for storm translation effects, the authors are encouraged to discuss the potential differences in results had the older Holland (1980) model been used in the current work. This comparison could provide valuable insights into the model's influence on the study's outcomes.**

Response:

Thank you for your insightful comment and kind suggestion. The Holland (2008) model refined the use of the Holland B parameter, which significantly affects the shape of the pressure gradient and wind speed profile. By adjusting this parameter, the model can better simulate the wind field characteristics of different typhoons. We compared the simulation result of the two models, as shown in the following figure, it appears that the water level simulation of Holland (1980) may be a more suitable model for the case of Typhoon Nida than the Holland (2008) model.

[Figure]

**Figure** Time series comparisons of measured and modeled water levels at (a) Chiwan gauge (b) Hong Kong gauge (c) Guangzhou gauge

**4. The font size in certain figures, such as the legend in the subplots on the right-hand side of Figure 4 and the tick labels on the x- and y-axes in Figures 7 through 9, is too small and may be difficult for readers to interpret.**

Response:

We are grateful for your constructive proposal and have made the necessary corrections to the right-hand label of Figure 4 in the manuscript, also has shown in Fig 1. Additionally, we have adjusted the font size of the tick labels in Figures 6 to 9 as outlined below.

[Figure]

**Figure 6.** Time series of the nonlinear components of Typhoon Nida at P1, P2, P3, P4 and P5 in *x* direction (left) and *y* direction (right)

[Figure]

**Figure 7.** Time series of the nonlinear components at P1, P2, P3, P4 and P5 in x direction (left) and y direction (right) when maximum storm surge coincides with the HLW tidal phase

[Figure]

**Figure 8.** Time series of the nonlinear components at P1, P2, P3, P4 and P5 in x direction (left) and y direction (right) when the maximum storm surge coincides with the LHW tidal phase

[Figure]

**Figure 9.** Time series of the nonlinear components at P1, P2, P3, P4 and P5 in *x* direction (left) and *y* direction (right) when the maximum storm surge coincides with the LLW tidal phase.